# Early summer hydroclimatic signals are well captured by tree-ring earlywood width in the eastern Qinling Mountains, central China

Yesi Zhao[1,2], Jiangfeng Shi[1,3], Shiyuan Shi[1], Xiaoqi Ma[1], Weijie Zhang[1], Bowen Wang[1], Xuguang Sun[4], Huayu Lu[1], Achim Bräuning[2]

[1] School of Geography and Ocean Science, Nanjing University, Nanjing 210023, China
[2] Institute of Geography, Friedrich-Alexander-University Erlangen-Nürnberg, Erlangen 91058, Germany
[3] Laboratory of Tree-Ring Research, University of Arizona, Tucson 85721, USA
[4] School of Atmospheric Sciences, Nanjing University, Nanjing 210023, China

*Correspondence to*: Jiangfeng Shi (shijf@nju.edu.cn)

**Abstract.** In the humid and semi-humid regions of China, tree-ring width (TRW) chronologies offer limited moisture-related climatic information. To gather additional climatic information, it would be interesting to explore the potentials of the intra-annul tree-ring width indices (i.e., the earlywood width (EWW) and latewood width (LWW)). To achieve this purpose, TRW, EWW and LWW were measured from the tree-ring samples of *Pinus tabulaeformis* originating from the semi-humid eastern Qinling Mountains, central China. Standard (STD) and signal-free (SSF) chronologies of all parameters  were created using

these detrending methods including (1) negative exponential functions combined with linear regression with negative (or zero) slope (NELR), (2) cubic smoothing splines with a 50 % frequency cutoff at 67 % of the series length (SP67), and (3) age-dependent splines with an initial stiffness of 50 years (SPA50). The results showed that EWW chronologies were significantly negatively correlated with temperature, but positively correlated with precipitation and soil moisture conditions during the current early growing season. By contrast, LWW and TRW chronologies had weaker relationships with these climatic factors.

The strongest climatic signal was detected for the EWW STD chronology detrended with the NELR method, explaining 50 % of the variance of the May–July self-calibrated Palmer Drought Severity Index (MJJ scPDSI) during the instrumental period 1953–2005. Based on this relationship, the MJJ scPDSI was reconstructed back to 1868 using a linear regression function. The reconstruction was validated by comparison with other hydroclimatic reconstructions and historical document records from adjacent regions. Our results highlighted the potentials of intra-annual tree-ring indices for reconstructing seasonal

hydroclimatic variations in humid and semi-humid regions of China. Furthermore, our reconstruction exhibited a strong in-phase relationship with a newly proposed East Asian summer monsoon index (EASMI) before the 1940s on the decadal and longer timescales, which may be due to the positive response of the local precipitation to EASMI. Nonetheless, the cause for the weakened relationship after the 1940s is complex, and cannot be solely attributed to the changing impacts of precipitation and temperature.

## 1 Introduction

Most of the existing tree-ring width (TRW) based hydroclimatic reconstructions in China were conducted for the regions located between the 200-to 600-mm annual precipitation isolines (Liu et al., 2018b), close to the northern fringe of Asian Summer Monsoon realm. Nonetheless, there exists still a small number of hydroclimate reconstructions in the core monsoon

region, for example from Southeast China (e.g., Cai et al., 2017; Chen et al., 2016a; Shi et al., 2015), North China (e.g., Chen et al., 2016b; Hughes et al., 1994; Lei et al., 2014; Liu et al., 2002), and from the Hengduan mountains in Southwest China (e.g., Fan et al., 2008; Fang et al., 2010b; Gou et al., 2013; Li et al., 2017). Since precipitation is spatially highly variable (Ding et al., 2013), hydroclimatic variations close to the monsoon boundary cannot completely represent those in the core monsoon region (Liu et al., 2018b). Thus, additional hydroclimatic reconstructions are needed from the center of the monsoon region.

Some TRW chronologies within the monsoon region showed weak or unstable hydroclimatic signals (e.g., Li et al., 2016; Shi et al., 2012; Wang et al., 2018), making them unsuitable to derive reliable reconstructions. By contrast, intra-annually resolved tree-ring width (i.e., earlywood width (EWW) and latewood width (LWW)) provided stronger hydroclimatic signals than TRW in some cases (Chen et al., 2012; Zhao et al., 2017a). This might be because the movement of the monsoon rain belt can lead to an uneven distribution of precipitation during the growing season (Jiang et al., 2006), thus causing restrictions of water

availability during parts of the growing season, and limiting the growth of certain intra-ring sectors rather than of total ring width (Liu et al., 2018a). EWW and LWW have been successfully applied for reconstructing long-term variations of seasonal rainfall (Hansen et al., 2017), drought (standardized precipitation evapotranspiration index SPEI; Zhao et al., 2017b), and streamflow (Guan et al., 2018) in the monsoon region.

The eastern Qinling Mountains are located within the core region of the East Asian summer monsoon (EASM), and are

characterized by a transitional climate from warm-temperate to subtropical. In this region, Shi et al. (2012) built four TRW chronologies of *Pinus tabulaeformis* along an elevation gradient from 1200 to 1950 m above sea level (a.s.l.). The TRW chronologies of the two low-altitude sites, i.e., Baiyunshan and Longchiman, exhibited a positive response to precipitation and negative response to temperature during early summer, showing the sign of water stress. However, the dendroclimatic potentials of EWW and LWW were not explored.

Meanwhile, tree-growth at an adjacent site was found to be more restricted by the drought index, represented by the well-known PDSI (Palmer Drought Severity Index), rather than by precipitation or temperature (Peng et al., 2014). Briefly, the PDSI monitors the cumulative departure of surface water balance in terms of the difference between the amount of precipitation required to retain a normal water-balance level and the amount of actual precipitation (Palmer, 1965; Wells et al., 2004). In addition, the SPEI, a drought index representing water balance in the form of difference between precipitation and

potential evapotranspiration (Vicente-Serrano et al., 2010), was reported to constrain latewood growth at a well-drained site in the core region of EASM (Zhao et al., 2017a, b). According to these findings, complex drought indices like the PDSI and SPEI should also be incorporated into the climate-tree-growth relationship assessment of tree-ring variables.

Recently, a new East Asian summer monsoon index (EASMI) based on the 200 hPa zonal wind anomalies was introduced by Zhao et al. (2015). The new EASMI performs better in describing precipitation and temperature variations over East Asia than existing indices. Suppose a strong EASM occurred, the new EASMI would indicate abundant precipitation and relatively low temperature over the core region of EASM, particularly around the area of mei-yu/changma/baiu rainband (27.5°–32.5° N, 105°–120° E and 30°–37.5° N, 127.5°–150° E; Fig. 1a), which is the center of the leading mode of EASM precipitation (Wang et al., 2008; Zhao et al., 2015). It would be interesting to study the response of local hydroclimate to EASM from a long-term perspective in combination with these tree-ring materials and the new EASMI.

Considering the TRW series of *P. tabulaeformis* in Baiyunshan and Longchiman are mainly restricted by the early summer moisture conditions, we hypothesize that the early summer hydroclimatic signals might be strengthened only using EWW, since earlywood is mostly formed before mid-July (Zhang et al., 1982). Therefore, the objectives of this study are (1) to verify that EWW is more sensitive to early summer hydroclimatic factors than TRW and LWW for *P. tabulaeformis* at Baiyunshan and Longchiman, (2) to reconstruct early summer hydroclimate variations using EWW, and (3) to tentatively explore the relationship between the reconstructed hydroclimate variability and EASMI.

## 2 Materials and Methods

### 2.1 Study sites

In this study, the dated tree-ring samples of *P. tabulaeformis* are those presented by Shi et al. (2012). Briefly, they were collected from two sampling sites in Mount Funiu in 2006 and 2008 separately: Baiyunshan (33.63° N, 111.85° E) and Longchiman (33.68° N, 112.05° E; Fig. 1b). The sampling sites are located on mountain tops, where soils are thin and well-drained. The elevations of Baiyunshan and Longchiman range 1200–1300 m a.s.l., and 1340–1400 m a.s.l., respectively. The regional annual mean temperature and annual total precipitation are 14.1 °C and 809.1 mm, respectively. Most of the annual total precipitation drops in the warm season (Fig. 2). More detailed information of the study sites can be found in Shi et al. (2012).

### 2.2 Tree-ring data

*P. tabulaeformis* is a widely distributed conifer species in North China with the extension from 31° 00′ N to 43° 33′ N, and 103° 20′ E to 124° 45′ E (Xu et al., 1981). By studying the cambial dynamics of *P. tabulaeformis* in its northern distribution limit (43° 14.11' N, 116° 23.60' E, 1363 m a.s.l.), Liang et al. (2009) reported that the cell division in the cambial zone would start within the third week of May and complete by mid-September. A separate study in Northwest China (37° 02′ N, 104° 28′ E, 2456 m a.s.l.) found that the cambial cells of mature *P. tabulaeformis* would start the cell division in late spring and cease in late July/early August (Zeng et al., 2018). Since our sampling sites are located at lower latitudes, the cambial activity of *P. tabulaeformis* in our study area may start earlier and end later than those found in above studies (following the temperature-controlled phenology theory; Chen and Xu, 2012).

*P. tabulaeformis* generally exhibits an abrupt transition from light-colored earlywood to dark-colored latewood (Fig. S1; Liang and Eckstein, 2006). Based on the characteristics of the tree-ring anatomy, the earlywood and latewood segments of annual growth rings can be distinguished visually by the sudden change in cell size, lumen size, and color (Stahle et al., 2009). However, gradual transitions could occur in a few samples, making the earlywood-latewood boundary difficult to differentiate.

Therefore, only samples with distinct earlywood and latewood segments were used for subsequent measurements (Knapp et al., 2016). In total, 20 cores from 11 trees and 42 cores from 22 trees were selected from Baiyunshan and Longchiman, respectively. EWW and LWW were then measured using a LINTAB5 system at a resolution of 0.001 mm, and TRW was obtained by adding EWW and LWW of the same calendar years together.

## 2.3 Development of tree-ring width chronologies

Non-climatic growth trends need to be fitted and removed from each "raw" (untreated) EWW, LWW and TRW series, which is known as detrending (Cook et al., 1990). To check the effects of detrending methods on the preservation of climatic signals, three detrending methods were selected for comparison. These included were negative exponential functions combined with linear regression with negative (or zero) slope (NELR), cubic smoothing splines with a 50 % frequency cutoff of 67 % of the series length (SP67), and age-dependent splines with an initial stiffness of 50 years (SPA50). NELR is a deterministic method

based on the assumption that tree radial growth declines monotonically (Cook et al., 1990). SP67 has a good ability in fitting the potential low-and middle-frequency perturbations contained in ring-width series (Cook et al., 1990). It allows no more than half of the amplitude of variations with wavelength of two-thirds of the length of series being preserved in resulting indices (Melvin et al., 2007). SPA50 specifies annually varying 50 % frequency cutoff parameter for each year by adding the initial stiffness with ring age. Comparing to SP67, it makes the resulting spline become more flexible in the early years and

progressively stiffer in later years (Melvin et al., 2007). All raw ring-width series were divided by the estimated growth trends, and the resulting detrended ring-width series were averaged to generate the standard (STD) chronologies using the bi-weight robust mean method (Fig. S2). Since the traditionally fitted curves may contain low-frequency climatic signals (known as "trend distortion" problem; Melvin and Briffa, 2008), the signal-free (SSF) method was introduced to create a fitted growth curve without climatic signals by dividing the raw ring-width series by the STD chronology by means of iterations (Melvin

and Briffa, 2008). Therefore, we also developed SSF chronologies for climate analysis (Fig. S3). Following the methods described in Osborn et al. (1997), the variance of each chronology was stabilized to minimize the effects of sampling depth. The temporal extension for all width chronologies in Baiyunshan and Longchiman covers the periods 1841–2005 and 1850–2005, respectively. All the above processes were performed using the program RCSsigFree Version 45_v2b (Melvin and Briffa, 2008;http://www.ldeo.columbia.edu/tree-ring-laboratory/resources/software). Since the number of cores per tree in our study

was unequal, the signal of each chronology was estimated using the effective chronology signal ($Rbar_{eff}$) which incorporates the effective number of cores, within- and between-tree signals (Briffa and Jones, 1990). Besides, the expressed population signal (EPS), a function of $Rbar_{eff}$ and the number of trees, was applied to evaluate how well the sample chronology represents a theoretical chronology (Briffa and Jones, 1990; Wigley et al., 1984). The running $Rbar_{eff}$ and EPS for each chronology were

calculated over a 51-year window with a 50-year overlap using the function "rwi.stats.running" in R package "dplR" version 1.6.9 (Bunn et al., 2018). The minimum number of common years in any pair of ring-width series required for their correlation was set to 30 (Briffa and Jones, 1990). The reliable period for each chronology was determined based on the widely adopted EPS threshold value of 0.85 (Wigley et al., 1984).

5 The width chronologies from the two sampling sites show high degree of coherence as evidenced by their significant positive correlations ($p < 0.001$) during their common period 1850–2005 (Table S1). Moreover, the positive correlations remain significant ($p < 0.001$) after removing the influence of autocorrelations and linear trends from the tree-ring data (Table S1). These indicate that the two sites share common climatic signals. Therefore, we pooled all raw ring-width series from the two sites, and developed composite STD and SSF chronologies for EWW, LWW and TRW using the three detrending methods as 10 described above (Fig. S4). Statistics for each chronology including the starting year when EPS ≥ 0.85, standard deviation, mean sensitivity and first-order correlation coefficient (AR1) are shown in Table S2. In addition, several statistics were calculated to assess the degree of similarity among the detrended ring-width series over the common period 1915–2005 (Table S3). These statistics are variance explained by the first eigenvector ($Var_{pc1}$), $Rbar_{eff}$, signal-to-noise ratio (SNR), and EPS (Briffa and Jones, 1990; Trouet et al., 2006).

15 **2.4 Climate data**

Monthly mean maximum (Tmax), minimum (Tmin) and mean temperature (Tmean), and monthly total precipitation (Pre) were gathered from four nearby meteorological stations (Table 1; Fig. 1b). These climate data were obtained from the China Meteorological Administration. Regional temperature values were calculated by averaging the temperature time series from the four stations over their common period 1957–2005. Regional precipitation series were calculated by firstly deriving the 20 regional averages in terms of percentages, then multiplying the regional mean to transform the resulting series back to millimetre units (Jones and Hulme, 1996). The self-calibrated PDSI (scPDSI) and SPEI were also chosen as hydroclimatic factors. Here we used the scPDSI instead of PDSI because it has solved the PDSI problems in spatial comparisons by calculating the duration factors (weighting coefficients for the current moisture anomaly and the previous drought severity) based on the characteristics of the climate at a given location (Wells et al., 2004). The regional scPDSI was calculated by 25 averaging the CRU (Climate Research Unit) scPDSI grids (van der Schrier et al., 2013) over the area between 32° N to 34.5° N and 111° E to 112° E (Fig. 1b) where the meteorological stations utilized by CRU dataset were concentrated (Fig. S5; Table S4). The time span of scPDSI was selected as 1953–2005, because the stations used by CRU have a common period starting from July 1952 (Table S4). The SPEI has multi-timescales (Vicente-Serrano et al., 2010). To evaluate the influence of SPEI on monthly, seasonal and annual timescales, we calculated the regional SPEI on three timescales (1-month, 3-month and 12-30 month) using the R package "SPEI" version 1.7 (Beguería and Vicente-Serrano, 2017). The climatic factors used in SPEI calculation were regional Tmax, Tmin and Pre, which were derived from the four stations mentioned in Table 1. The time span of SPEI is 1957–2005.

In order to validate the reconstruction, we compared it with several hydroclimate time series and historical document records (Table 2). They were (1) the June–August PDSI from the No. 370 grid point of the Monsoon Asia Drought Atlas (MADA) at 33.75° N, 111.25° E over the period 1868–2005 (PDSI$_{Cook}$; Cook et al., 2010); (2) the dryness/wetness index (DWI) from the grid point at 33.75° N, 111.25° E over the period 1868–2000 (DWI$_{Yang}$; Yang et al., 2013); (3) reconstructed April–June precipitation based on TRW in Mount Hua over the period 1868–2005 (Pre$_{Chen}$; Chen et al., 2016b); and (4) drought/wet events recorded in historical documents over the period 1868–2005 (He, 1980; Wen, 2006). The DWI dataset was reconstructed from the historical documents and modern instrumental May–September precipitation in 120 sites over China (Chinese Academy of Meteorological Sciences, 1981). The dataset classified the degree of dryness and wetness into five grades: very wet (grade 1), wet (grade 2), normal (grade 3), dry (grade 4), and very dry (grade 5). Yang et al. (2013) has interpolated the DWI dataset into 2.5° latitude/longitude grid cells.

The EASM circulation was represented using a newly defined EASMI based on the 200 hPa zonal wind anomalies which was less affected by complex weather processes near the surface (Zhao et al., 2015). It was computed using:

$$\text{EASMI} = \text{Nor}[u(2.5°–10° N, 105°–140° E) − u(17.5°–22.5° N, 105°–140° E) + u(30°–37.5° N, 105°–140° E)] \quad (1)$$

where Nor and u are standardization and mean 200 hPa zonal wind, respectively. To understand the possible impacts of local precipitation and temperature (32° –34.5° N and 111° –112° E) on the relationship between the scPDSI and EASMI, the precipitation and temperature data were extracted from the gridded precipitation dataset Global Precipitation Climatology Centre Version 7 (GPCC v7; Schneider et al., 2015), and gridded temperature dataset Climatic Research Unit Time-Series Version 4.01 (CRU TS 4.01; Harris et al., 2014), respectively. The gridded dataset can represent the variations of precipitation and temperature over East China in the 20$^{th}$ century (Wang and Wang, 2017; Wen et al., 2006).

**2.5 Statistical methods**

To investigate the climate response of different tree-ring parameters (EWW, LWW, and TRW), we first calculated the Pearson correlation coefficients of the STD and SSF tree-ring width chronologies using monthly climate time series. The time window for the correlation analysis spanned from January of two years before tree-ring formation to October of the current growth year. Next, correlations were calculated between the prewhitened and linearly detrended chronologies and climate time series in order to evaluate the possible effects of autocorrelations and secular trends. The prewhitening procedure was performed using the "ar" function in R package "stats" version 3.5.1 (R Core Team, 2018). The appropriate autoregressive order was automatically determined by the Akaike Information Criterion (Akaike, 1974). The linear detrending procedure was performed based on the "detrend" function in Matlab R2016a (The MathWorks, Inc., 2016). To find the strongest climate-growth relationship, we analyzed the response of different tree-ring parameters to multi-month averaged scPDSI (which had the stronger impacts on tree-growth than other climatic factors; see the results for details). Finally, we adopted the wavelet coherence method (Grinsted et al., 2004) to test the temporal stability and possible lags of the climate-growth relationship on different frequency domains.

A simple linear regression model was applied to establish the transfer function using May–July (MJJ) scPDSI as the predictand, and the NELR based EWW STD chronology as the predictor (which had the strongest relationship; see the results for detail) over the period 1953–2005. Temporal stability of the model was tested by splitting the MJJ scPDSI into two sub-periods (1953–1979 and 1979–2005) for calibration and verification. In the process, some statistics (including correlation coefficient

($r$), explained variance ($R^2$), reduction of error ($RE$), coefficient of efficiency ($CE$), and the sign-test) were applied (Meko and Graybill, 1995). Meanwhile, the possible autocorrelation and trend contained in the regression residuals were evaluated using the Durbin-Watson test ($DW$; Durbin and Watson, 1950) using the "dwtest" function in R package "Lmtest" version 0.9-36 (Zeileis and Hothorn, 2002), and the two-sided Cox and Stuart trend test ($CS$; Cox and Stuart, 1955) using the R package "snpar" version 1.0 (Qiu, 2014), respectively. A $DW$ value of 2 means no first order autocorrelation in the residuals, whereas

values larger (less) than 2 indicate for negative (positive) autocorrelation. The $DW$ test has the null hypothesis that the autocorrelation of the residuals is 0. The two-sided $CS$ trend test has the null hypothesis that there is no monotonic trend in the residuals. The variance of the MJJ scPDSI reconstruction was adjusted to match the variance of instrumental MJJ scPDSI during the calibration period using Equation (2),

$$\text{Adj\_Rec}_i = \frac{(\text{Rec}_i - \overline{\text{Rec}_{cal}})}{\sigma(\text{Rec}_{cal})} \times \sigma(\text{Ins}_{cal}) + \overline{\text{Ins}_{cal}} \qquad (2)$$

where, the $\text{Rec}_i$ and $\text{Adj\_Rec}_i$ are the reconstructed value and its variance adjusted value for a specific year $i$, respectively. The $\overline{\text{Rec}_{cal}}$ and $\overline{\text{Ins}_{cal}}$ are the arithmetic mean of the reconstructed and instrumental values during the calibration period (it is 1953–2005 in this study), respectively. The $\sigma(\text{Rec}_{cal})$ and $\sigma(\text{Ins}_{cal})$ are the corresponding standard deviations.

To evaluate the spatial representativeness of our reconstruction, spatial correlations were calculated between the reconstructed MJJ scPDSI and CRU scPDSI 3.25 dataset (van der Schrier et al., 2013) using the KNMI Climate Explorer

(http://climexp.knmi.nl/start.cgi). For comparison purpose, all the hydroclimatic reconstructions were divided into interannual (< 10 years), and decadal and longer-term components (> 10 years), respectively. Decadal and longer frequency components were derived by lowpass filtering of the original reconstructions using the the adaptive 10 point "Butterworth" low-pass filter at 0.1 cut-off frequency (Mann, 2008). Interannual frequency components were obtained by subtracting the decadal and longer-term components from the original reconstructions. The low-pass filtering technique is capable to preserve trends near time

series boundaries (Mann, 2008).

Following the definition of Zhao et al. (2015), we calculated the MJJ EASMI using the 200 hPa zonal wind dataset. The dataset covering the period 1868-2005 was obtained from the National Oceanic and Atmospheric Administration-Cooperative Institute for Research in Environmental Sciences Twentieth Century Reanalysis V2c (NOAA-20C; Compo et al., 2011). The relationship between EASMI and our reconstruction was firstly evaluated using the wavelet coherence method (Grinsted et al.,

2004). To explore the connections of precipitation and temperature with scPDSI and EASMI, 21-year moving window correlation analyses were conducted between the decadal-filtered MJJ EASMI, reconstructed scPDSI, local precipitation and temperature. Moreover, empirical orthogonal function (EOF) analysis and spatial correlation analysis were performed to assess the impacts of the changed leading EASM mode on the relationship between decadal-filtered EASMI and local precipitation.

The filtering procedure was conducted using the "Butterworth" low-pass filter (Mann, 2008) as mentioned above. The filtering, EOF and correlation analyses were performed in Matlab R2016a (The MathWorks, Inc., 2016) and the plots were drawn with Surfer 10 (Golden Software, LCC, 2011).

The significance tests for all observed correlation coefficients were conducted using Monte Carlo method (Efron and Tibshirani, 1986). In detail, modelled time series with the same structure as the original series were produced in accordance to the frequency domain method of Ebisuzaki (1997). Then, correlation coefficients were computed between the modelled time series. The above processes were repeated 1000 times to obtain 1000 modelled correlation coefficients. The significance threshold was estimated based on the probability distribution of the modelled 1000 correlation coefficients. The procedure was performed using the algorithms of Macias-Fauria et al. (2012).

## 3 Results and Discussion

### 3.1 Stronger hydroclimatic signals derived from EWW

The EWW chronologies generated using different detrending and standardization methods were significantly negatively correlated with Tmax, Tmean during May–June; and significantly positively correlated with precipitation in May (Fig. 3). By relating to drought indices, all EWW chronologies were significantly positively correlated with the 1-month SPEI in May; 3-month SPEI during May–June; 12-month SPEI during May–October; and scPDSI during April–October. Particularly, EWW showed a much longer-term response to the multi-month SPEI and scPDSI than to precipitation after May. This may be because the summer temperatures affected the soil water status as reflected by their negative correlations with EWW. Besides, soil has a memory effect for previous drought conditions, and this effect is considered in the multi-month SPEI and scPDSI (Vicente-Serrano et al., 2010; Dai, 2011). The scPDSI had higher correlation with tree-ring width than the SPEI. This indicates that the scPDSI is more suitable for monitoring the influence of soil moisture status on tree growth at our sampling sites, however the reasons for this remain unknown at the moment. The significant correlations between EWW and drought indices during autumn should not be regarded as a real drought impact, as the earlywood growth would terminate in the mid- and late-growing season (Larson, 1969). For LWW, during the current growing season, the highest correlation was found between the NELR based LWW STD and July scPDSI ($r = 0.37$, $p < 0.01$). However, this correlation was much lower than that between the NELR based EWW STD and the July scPDSI ($r = 0.62$, $p < 0.01$), indicating that LWW was less sensitive to the scPDSI. TRW generally exhibited a similar climate response as EWW but with relatively lower correlations. Taking the NELR based STD chronologies as an example, the correlation coefficients between TRW and the monthly scPDSI from May to July were 0.59 ($p < 0.01$), 0.58 ($p < 0.01$) and 0.58 ($p < 0.01$), respectively. However, for EWW, these correlation coefficients were 0.66 ($p < 0.01$), 0.66 ($p < 0.01$) and 0.62 ($p < 0.01$), respectively. Similar response patterns were also revealed by the correlation coefficients between the prewhitened and linearly detrended series (Fig. 4), indicating that autocorrelations and secular trends in the tree-ring width chronologies and climate time series had limited effects on the relationships with climate.

Significant climate-growth relationships were also observed with months prior to the current growing season. For example, most of EWW and LWW chronologies exhibited negative response to Tmax and Tmean during the late summer and early autumn of the last year (Figs. 3 and 4). High temperatures in the late growing season of last year may enhance soil water evaporation, thus inducing moisture stress and limiting the accumulation of photosynthetic products available for the next

year's tree-growth (Peng et al., 2014). The influence of moisture status prior to the current growing season is also reflected by the significant positive correlations between LWW and drought indices from September of two years earlier to May of the previous year. However, EWW had lower correlations with these monthly drought indices. A plausible explanation may be that the interannual variations of EWW were mainly contributed by the moisture status of the growth year.

Since the impacts of scPDSI on tree growth can last for several months, we analyzed the responses of various tree-ring width

parameters to the multi-month averaged scPDSI. The strongest climate-growth relationship was found between the NELR based EWW STD chronology and the MJJ scPDSI ($r = 0.707$; $p < 0.01$; Fig. S6). Meanwhile, correlation coefficients derived from the methods SP67 and SPA50 were 0.67 ($p < 0.01$) and 0.68 ($p < 0.01$), respectively, which were lower than those based the on NELR method (Fig. S6). This may be because the downward trend in MJJ scPDSI was better preserved using the NELR detrending method (Fig. S7). In addition, the correlation coefficient between the NELR based EWW SSF chronology and the

MJJ scPDSI was 0.705 ($p < 0.01$), which was quite close to that using the traditional STD method, indicating that the effects of so-called "trend distortion" in our tree-ring series were small.

We further tested the temporal stability and possible lags (leads) in the relationships between NELR based STD chronologies and the MJJ scPDSI on different frequency domains (Fig. 5). EWW generally has high degrees of coherence with the MJJ scPDSI on all timescales (2- to 18-year), except for the periodicities between 3.5- and 6.5-year. By contrast, LWW only varied

in-phase with the MJJ scPDSI, but with some lags during the period from the 1970s to the 1990s on the timescales shorter than 12 years. Moreover, LWW was inversely correlated with the MJJ scPDSI during the 1960s in the periodicities of 4- to 6-year. TRW showed an unstable relationship and certain lags to the MJJ scPDSI in the periodicities of 6- to 11-year. Therefore, it can be concluded that EWW has more stable relationships with the MJJ scPDSI than LWW and TRW.

Previous studies based on TRW demonstrated that moisture status of the current growing season could strongly affect the

radial growth of *P. tabulaeformis* (e.g., Cai and Liu, 2013; Cai et al., 2014; Cai et al., 2015; Chen et al., 2014; Fang et al., 2010a; Fang et al., 2012b; Li et al., 2007; Liang et al., 2007; Liu et al., 2017; Song and Liu, 2011; Sun et al., 2012). Fast radial growth of *P. tabulaeformis* usually occurs during the early growing season (Liang et al., 2009; Shi et al., 2008; Zeng et al., 2018). Increased water deficiency due to the rising temperature and inadequate rainfall in the early growing season induces water stress, suppressing cell division and expansion (Fritts, 1976), and resulting in the formation of narrow earlywood bands.

The reduced sensitivity of LWW to the moisture status of the current growing season may be due to that ample water supply in the rainy season (July–August; Fig. 2). TRW and EWW shared similar climatic response. This is because EWW represents the majority of TRW (on average, the portion of EWW of TRW accounts for 65.8%). However, since TRW is also contributed by LWW, the response of TRW to moisture status in the current growing season was somewhat weaker than EWW.

**3.2 MJJ scPDSI reconstruction using NELR based EWW STD chronology**

From above analyses, we identified the MJJ scPDSI as the target for hydroclimate reconstruction; and the NELR based EWW STD chronology as the predictor (Fig. 6a). The transfer function was estimated using a simple linear regression model, as expressed below:

$\text{MJJ scPDSI} = 4.74\text{EWW} - 4.32; (R^2 = 0.5, n = 53, p < 0.001)$ ,    (3)

The model explained 50 % of the actual MJJ scPDSI variance over the period 1953–2005. The calibration-verification tests revealed that $r$, $R^2$ and the sign-test were significant at the 0.01 level, and that $RE$ and $CE$ values were positive (Table 3). In addition, the $p$-value generated from $DW$ and $CS$ tests were above 0.05, indicating that there were neither autocorrelation nor long-term trends in the regression residuals (Table 3; Fig. 6b). All test results confirmed that the model is valid (Cook et al.,
1999; Fritts, 1976).

Based on Equation (3), we reconstructed MJJ scPDSI of the study region back to 1868 (Fig. 6c). We adjusted the variance of the reconstruction to match the variance of instrumental MJJ scPDSI during the calibration period (1953–2005). Spatial correlation analysis indicated that the reconstruction most strongly represents Central China, including the western part of Henan, northern part of Hubei, and southern part of Shaanxi provinces (Fig. 1a).

**3.3 Comparing the reconstructed MJJ scPDSI with other climate reconstructions and historical records**

On the interannual timescale (Fig. 7a–c), our reconstruction is significantly correlated with the $\text{PDSI}_{\text{Cook}}$ ($r = 0.37$; $p < 0.01$; 1868–2005), and $\text{Pre}_{\text{Chen}}$ ($r = 0.52$; $p < 0.01$; 1868–2005). On the decadal and longer timescales, our reconstruction is significantly correlated with all other reconstructions from the study region (Fig. 7e and f). The common drought periods occurring in the 1870s and the 1920s in North and West China (Cai et al., 2014; Chen et al., 2014; Fang et al., 2012a; Kang et
al., 2013; Liang et al., 2006; Liu et al., 2017; Zhang et al., 2017) were also reflected in our reconstruction.

However, it should be noted that our reconstruction has also some mismatches with others. On the interannual timescale, our reconstruction is not significantly correlated with the $\text{DWI}_{\text{Yang}}$ over the whole period 1868–2000 ($r = -0.06$; $p = 0.64$; Fig. 7b). This is probably due to the limited ability of historical documents to capture high-frequency climatic variations (Zheng et al., 2014). On the decadal and longer timescales, our reconstruction varied out-of-phase with $\text{PDSI}_{\text{Cook}}$ during the period from the
late 1940s to the early 1960s (Fig. 7d). It was only weakly correlated with $\text{DWI}_{\text{Yang}}$ after the 1940s (Fig. 7e), and led $\text{Pre}_{\text{Chen}}$ during the period of 1900s–1930s (Fig. 7f). The might be due to several reasons. Firstly, the reconstructions focused on different target seasons (June–August for $\text{PDSI}_{\text{Cook}}$, May–September for $\text{DWI}_{\text{Yang}}$, and April–June for $\text{Pre}_{\text{Chen}}$ which is before the rainy season). Secondly, the $\text{DWI}_{\text{Yang}}$ after the 1940s was calculated using instrumental May–September precipitation and the chronology of Chen et al. (2016b) also reflects precipitation, while the scPDSI is influenced not only by precipitation but
also temperature and previous drought conditions. Thirdly, in the MADA network includes only a limited number of tree-ring sites in our study region, which may cause deviations on the local scale.

We also compared the dry and wet events derived from our reconstruction with historical records. Following Palmer (1965), the moderately to severely dry (wet) events are defined based on the scPDSI values less than –2 (larger than 2). Interestingly, all dry events and 70 % of the wet events in our reconstruction can be verified by corresponding descriptions in historical records (Table 4). However, there are still some mismatches between our reconstruction and the historical records. For example,

no relevant record is found for the year 1983 when an extreme wet event is shown in our reconstruction. Meanwhile, some historical events are not reflected in our reconstruction, such as the wet event in 1963 and the dry event in 1942 (Wen, 2006). In addition, our reconstruction generally demonstrates a wetter condition compared with other series (especially in its early part), and fails to capture the extreme drought years documented in the past (e.g., 1976–1978 and 1914; Wen, 2006). This might be because the low-frequency hydroclimatic variability preserved in EWW was entangled with the declining biological

trend, and was partly removed during the detrending process using simple curve fitting methods (Briffa et al., 1996). A possible approach to preserving the low-frequency hydroclimatic signals is to adopt the Regional Curve Standardization (RCS) method (Briffa et al., 1992), but this requires more field work in the future to collect additional tree-ring samples with an even distribution of different age classes (Briffa et al., 1996).

## 3.4 Connections with EASMI

In general, the reconstructed MJJ scPDSI and EASMI exhibit an in-phase relationship before the 1940s on the decadal and longer timescales (Fig. 8). This in-phase relationship was further verified after conducting a 21-year moving window correlation analysis on the decadal-filtered scPDSI and EASMI (Fig. 9a, b and e). Since EASM directly drives precipitation rather than the scPDSI, we compared the EASMI with the local precipitation (32° N to 34.5° N and 111° E to 112° E). We found that the local precipitation also exhibits similar variations as the scPDSI and EASMI before the 1940s (Fig. 9a–c).

Therefore, the in-phase relationship between the decadal-filtered EASMI and scPDSI before the 1940s may be due to the fact that a stronger EASM could enhance local precipitation, thus increasing the soil moisture content. Interestingly, the correlation between the local precipitation and EASMI weakened after the 1940s, and even became negative since the 1970s (Fig. 9e). We attribute this unstable relationship to the variable nature of EASMI. In fact, the EASMI was designed to capture the leading mode of EASM precipitation variability, whose largest loading is in general located at the mei-yu/changma/baiu rainband

(Zhao et al., 2015). In other words, the mei-yu/changma/baiu rainfall has the most robust relationship with the EASMI. Whilst the leading pattern of EASM precipitation could change on interdecadal timescale and longer timescales (which is still elusive due to the limited paleo-precipitation record), the unique importance of mei-yu/changma/baiu in EASM most likely remains (Wang et al., 2008). With the changing precipitation pattern of EASM, the precipitation outside of the mei-yu/changma/baiu rainband could be in-phase, out-of-phase and uncorrelated with mei-yu/changma/baiu rainfall (Wang et al., 2008), which

would manifest an unstable relationship with EASMI. The EASM experienced an abrupt shift in the late 1970s, which caused a change of the leading mode of EASM precipitation (Wang, 2001; Ding et al., 2008). We demonstrated how this mode change affects the relationship between EASMI and precipitation in the eastern Qinling Mountains. As shown in Fig. 10a, the anomalies of the decadal-filtered MJJ precipitation exhibited similar variations over the Yangtze River basin and Yellow-

Huaihe River basins during 1901–1978. However, they were divided by the Yangtze River, showing a dipole pattern during 1979–2005 (Fig. 10b). During both periods, the loading centres were located south of Yangtze River basin (27°–30° N), and the decadal-filtered MJJ precipitation in this area was well captured by the designed EASMI as manifested by their significant positive correlations ($p < 0.1$; Fig. 10c, d). On the contrary, the decadal-filtered MJJ precipitation north of Yangtze River (including our sampling sites) varied out-of-phase with that south of Yangtze River basin after the late 1970s, thus being negatively correlated with the EASMI.

The weakened scPDSI-EASMI relationship after the 1940s cannot be solely attributed to the change of the EASM precipitation mode, because the scPDSI also showed a weakened relationship with the local precipitation simultaneously (Fig. 10f). Neither can it be ascribed simply by the fact that the variations of scPDSI became dominated by temperature either, as no enhanced scPDSI-temperature relationship was found (Fig. 10f). This may be because the scPDSI is not a simple formula based on precipitation and temperature, but a complex function incorporating previous drought conditions and current moisture departure (Wells et al., 2004). In addition to precipitation and temperature, the available energy, humidity and wind speed can also affect the scPDSI via controlling evapotranspiration (Sheffield et al., 2012). Therefore, a variety of climate data is required to identify the specific cause for the observed weakened relationship. However, an exact determination of the causing factors is difficult due to the limited existing climate records.

**4 Conclusions**

Besides TRW, climatic responses of EWW and LWW were explored for the tree-ring samples of *P. tabulaeformis* from the eastern Qinling Mountains, Central China. Regardless of the detrending and standardisation methods used, the resulting EWW chronologies are more sensitive to early summer soil moisture conditions (scPDSI) than LWW and TRW during the instrumental period 1953–2005. The MJJ scPDSI (1868–2005) reconstructed from the NELR based EWW STD chronology captures the past early summer hydroclimatic fluctuations, which is further validated by other proxy-based reconstructions and historical document records from adjacent regions. This indicates that EWW has a great potential to reconstruct early summer hydroclimatic conditions in the study area. Moreover, on the decadal and longer timescales, our EWW-based hydroclimate reconstruction shows a strong in-phase relationship with the EASMI before the 1940s, which may be related to the positive response of the local precipitation to EASM intensity. Our finding differs from results developed at a well-drained site in South China, where strongest moisture signals are contained in LWW of a different tree species (Zhao et al., 2017a, b). Therefore, more EWW and LWW related studies should be conducted to reveal the tree- and site-specific climate signals in humid and semi-humid regions of China.

**Data availability**

The reconstructed May–July scPDSI was included in the supplementary material (Table S5). DWI, precipitation reconstruction, and dry/wet events recorded in historical documents are available from corresponding authors or publications. MADA is available from https://www.ncdc.noaa.gov/paleo-search/study/10435 (Cook et al., 2010). The 200 hPa zonal wind dataset of NOAA-20C is available from https://www.esrl.noaa.gov/psd/data/gridded/data.20thC_ReanV2c.html (Compo et al., 2011). The gridded dataset CRU scPDSI 3.25 is available from https://crudata.uea.ac.uk/cru/data/drought/ (van der Schrier et al., 2013). The gridded precipitation dataset GPCC v7 is available from https://opendata.dwd.de/climate_environment/GPCC/html/fulldata_v7_doi_download.html (Schneider et al., 2015). The gridded temperature dataset CRU TS 4.01 is available from https://crudata.uea.ac.uk/cru/data/hrg/cru_ts_4.01/cruts.1709081022.v4.01/tmp/ (Harris et al., 2014).

**Author contributions**

YZ and JS designed the study. JS provided the tree-ring samples. YZ performed tree-ring width measurement, data analyses and interpretation. JS, SS, XS and HL assisted in data interpretation. YZ wrote the first draft of the paper. All authors revised the paper.

**Competing interests**

The authors declare that they have no conflict of interest.

**Acknowledgments**

The study was supported by the Key R&D Program of China (Grant No. 2016YFA0600503), the National Natural Science Foundation of China (Grant No. 41671193), and the China Scholarship Council (Grant No. 201706190150 and 201806195033). We thank Dr. Feng Chen for providing his reconstructed precipitation data, and Mr. Zhou Yu for his help in tree-ring width measurements. We would also like to express our gratitude to EditSprings (https://www.editsprings.com/) for the expert linguistic services provided.

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

**Table 1.** Characteristics of climate data series used in correlation analyses.

| Climate data | Source | Longitude (° E) | Latitude (° N) | Elevation (m a.s.l.) | Temporal cover |
|---|---|---|---|---|---|
| Tmax, Tmean, Tmin, Pre | Luanchuan (LC) meteorological station | 111.6 | 33.8 | 750.3 | 1957–2005 |
|  | Xixia (XX) meteorological station | 111.5 | 33.3 | 250.3 | 1957–2005 |
|  | Ruyang (RY) meteorological station | 112.5 | 34.2 | 307.8 | 1957–2005 |
|  | Nanzhao (NZ) meteorogical station | 112.6 | 33.6 | 199.7 | 1956–2005 |
| scPDSI | CRU scPDSI 3.25 (van der Schrier et al., 2013) | 111–112 | 32–34.5 | — | 1953–2005 |
| SPEI | Calculated in R using the SPEI package with the monthly maximum and minimum temperature, and total precipitation (Beguería and Vicente-Serrano, 2012) | — | — | — | 1957–2005 |

**Table 2.** Long-term hydroclimatic reconstructions and East Asian summer monsoon index (EASMI) selected for comparing with the reconstructed MJJ scPDSI from this study.

| Time series | Source | Longitude (° E) | Latitude (° N) | Temporal cover |
|---|---|---|---|---|
| June–August PDSI | Monsoon Asia Drought Atlas (MADA; Cook et al., 2010) | 111.25 | 33.75 | 1868–2005 |
| April–June precipitation in Mount Huashan (HS) | Chen et al. (2016b) | 110.08 | 34.47 | 1868–2005 |
| Dryness/wetness index (DWI) | Yang et al. (2013) | 111.25 | 33.75 | 1868–2000 |
| EAMSI | Calculated using the 200 hPa zonal wind anomalies (NOAA-20C; Compo et al., 2011) according to the definition of Zhao et al. (2015) | — | — | 1868–2005 |

**Table 3.** Statistics of regression models for split calibration-verification periods.

| Calibration period | $r$ | $R^2$ | $DW$ value ($p$-value) | $CS$ $p$-value | Verification period | $RE$ | $CE$ | Sign-test |
|---|---|---|---|---|---|---|---|---|
| Full period (1953−2005) | 0.71** | 0.50** | 2.03 (0.53) | 0.85 | — | — | — | — |
| Early half (1953−1979) | 0.68** | 0.46** | 2.02 (0.49) | 1 | Late half (1979−2005) | 0.53 | 0.53 | 22+/5−** |
| Late half (1979−2005) | 0.73** | 0.54** | 2.02 (0.50) | 1 | Early half (1953−1979) | 0.46 | 0.45 | 21+/6−** |

** $p < 0.01$; $r$, Pearson correlation coefficient; $R^2$, explained variance; $DW$, Durbin-Watson test; $CS$, Cox and Stuart trend test; $RE$, reduction of error; and $CE$, coefficient of efficiency.

**Table 4.** Moderately to severely dry (scPDSI ≤ –2) and wet (scPDSI ≥ 2) events derived from the MJJ scPDSI reconstruction and corresponding descriptions from historical records.

| Event type | Year | scPDSI | Description |
|---|---|---|---|
| Dry | 1879 | –3.61 | A mega-drought occurred caused a great famine over Henan, Shaanxi and other provinces in North China in the early Guangxu reign (1876–1879)[a] |
| | 1900 | –2.24 | Severe drought from spring to Autumn over Henan and Shaanxi |
| | 1923 | –2.28 | Drought over Henan and Shaanxi |
| | 1926 | –2.33 | No harvest at Ruyang (West Henan) due to severe drought |
| | 1929 | –2.53 | Summer drought over Henan and Shaanxi |
| | 1994 | –2.12 | Severe drought occurred in April, May and July over west Henan |
| | 1995 | –2.10 | Intensified drought severity since April 22 over Henan |
| | 2000 | –2.94 | The drought from February to May was the worst one since 1950 over Henan |
| Wet | 1869 | 2.31 | Flood in summer and autumn over Henan |
| | 1883 | 2.62 | Persistent rainfall in summer at Shanxian and Mianchi (Northwest Henan) |
| | 1885 | 3.07 | Flood in summer at Lingbao and Shanxian (Northwest Henan) |
| | 1894 | 3.06 | Not available |
| | 1895 | 2.11 | Flood of the Qinhe River (Northwest Henan) in summer |
| | 1898 | 3.77 | Severe flood in summer at Lushi (Northwest Henan), Shangnan (Southeast Shaanxi) and Danjiang (Northwest Hubei) |
| | 1905 | 2.26 | Persistent rainfall in spring and summer over Henan |

| 1906 | 3.65 | Heavy rainfall in summer over Henan |
|------|------|-------------------------------------|
| 1910 | 2.05 | Flood in summer and autumn over Henan |
| 1911 | 3.84 | Heavy rainfall in summer over Henan |
| 1912 | 2.01 | Heavy rainfall and flood in summer over Nanyang (Southwest Henan) |
| 1933 | 2.51 | Heavy rainfall in summer over Henan and Shaanxi |
| 1934 | 3.11 | Summer rainfall over Henan, South Shaanxi and Northwest Hubei |
| 1936 | 3.72 | Not available |
| 1944 | 2.38 | Flood over Henan; Rainstorm in Zhenan (Southeast Shannxi) on July 8; The Tianhui Channel (Southeast Shaanxi) was destroyed by flood on May 13 |
| 1948 | 2.81 | Wheat loss caused by summer rainfall |
| 1949 | 2.95 | Not available |
| 1973 | 2.97 | Not available |
| 1980 | 2.07 | Rainfall in June was higher than usual for most regions over Henan |
| 1983 | 4.15 | Not available |
| 1984 | 2.33 | From June to September, there were 5 large-scale rainstorms over Henan |
| 1990 | 2.32 | Not available |

[a] Historical description of the 1879 drought event is cited from He (1980), and others from Wen (2006).

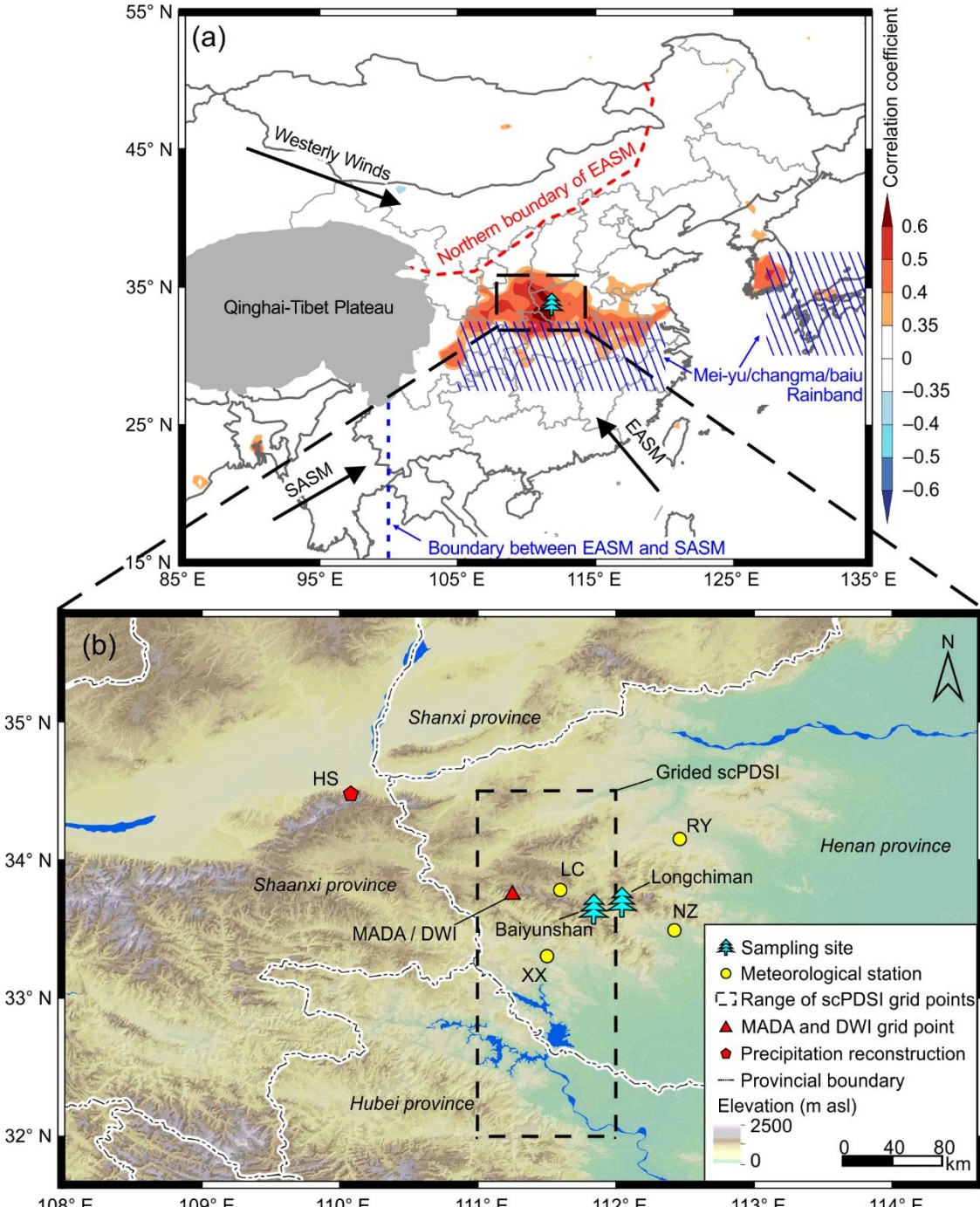

**Figure 1.** Map of the study region. **(a)** Location of the sampling site (tree symbol), and the spatial correlations between the

May–July (MJJ) scPDSI reconstruction and the gridded scPDSI dataset (van der Schrier et al., 2013) during the period 1953–

2005. The color bar indicates the correlation coefficient. The blue dashed line (100° E) indicates the boundary between East

Asian summer monsoon (EASM) and South Asian summer monsoon (SASM; Tang et al., 2010). The red dashed line indicates the northern boundary of EASM (Chen et al., 2018). The blue shaded area represents the mei-yu/changma/baiu rainband (Zhao et al., 2015). **(b)** Partial enlargement of the study region. The circles indicate the locations of the four meteorological stations (LC: Luanchuan, XX: Xixia, RY: Ruyang, and NZ: Nanzhao). The triangle indicates the location of selected grid data from the datasets Monsoon Asia Drought Atlas (MADA; Cook et al., 2010) and Dryness/Wetness Index (DWI; Yang et al., 2013). The pentagon indicates a tree-ring width based precipitation reconstruction in Huashan Mount (HS; Chen et al., 2016b). The dashed rectangle indicates the gridded scPDSI obtained from the gridded scPDSI dataset (van der Schrier et al., 2013).

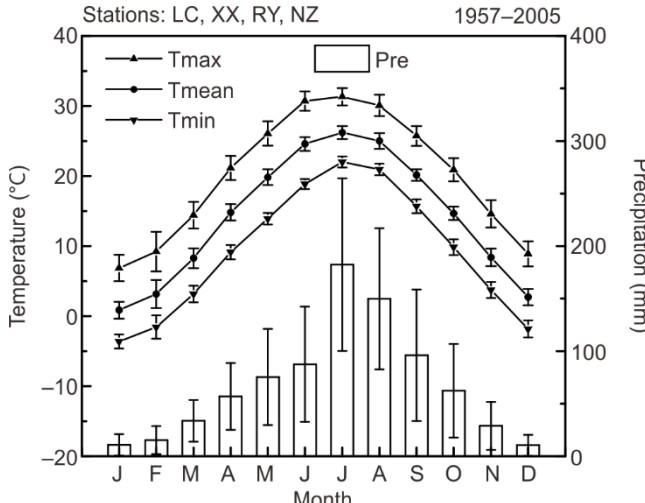

**Figure 2.** Monthly maximum, mean, minimum temperature (Tmax, Tmean, Tmin), and total precipitation (Pre) averaged from the four selected meteorological stations (LC: Luanchuan, XX: Xixia, RY: Ruyang, and NZ: Nanzhao) during the period 1957–2005. Error bar denotes ± one standard deviation.

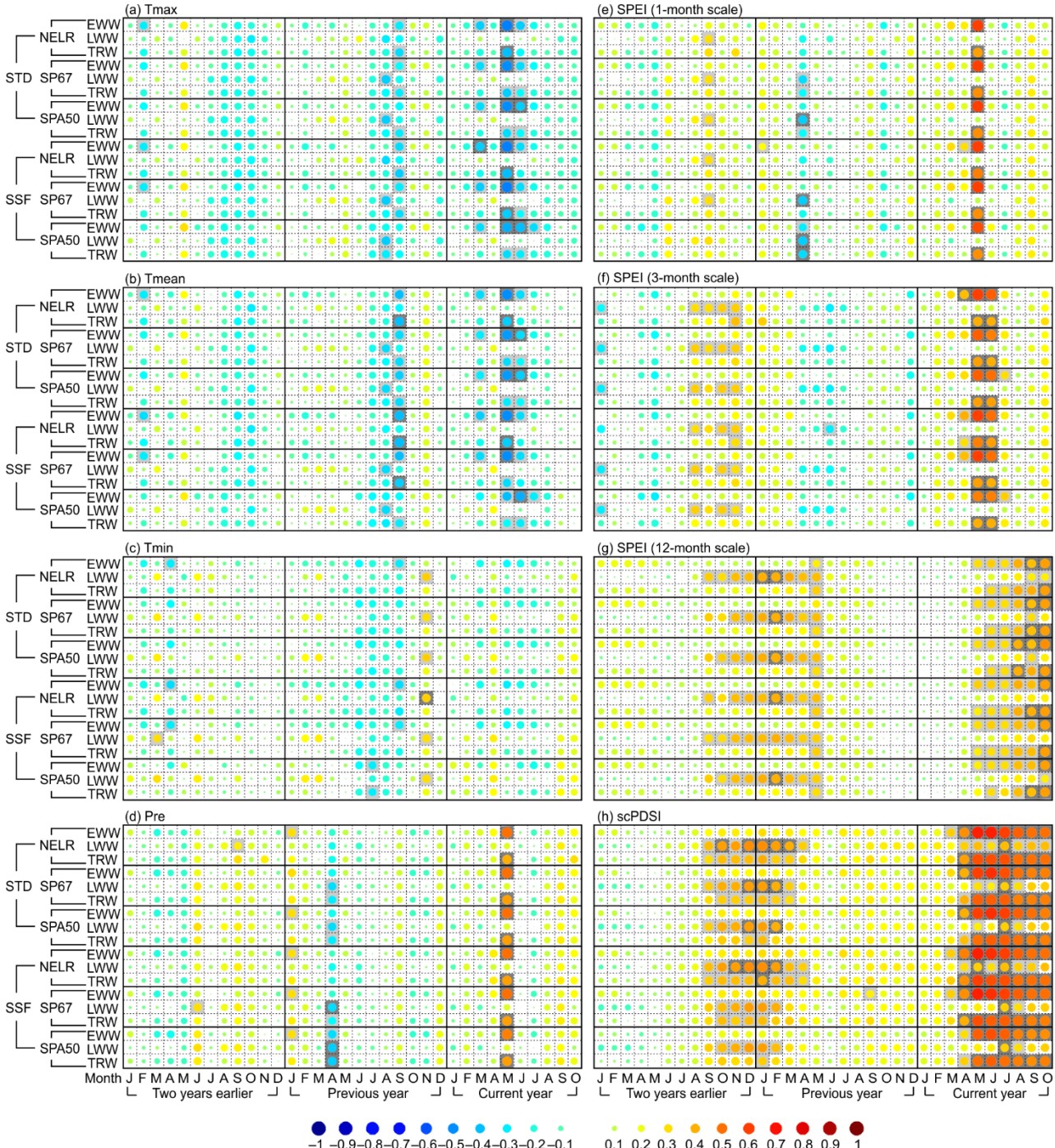

**Figure 3.** Matrix plots for the correlation coefficients between tree-ring width chronologies and monthly climate time series

from January of two years earlier to October of the current year. The climatic factors are monthly **(a)** maximum temperature

(Tmax), **(b)** mean temperature (Tmean), **(c)** minimum temperature (Tmin), **(d)** total precipitation (Pre), **(e)** SPEI of 1-month scale, **(f)** SPEI of 3-month scale, **(g)** SPEI of 12-month scale, and **(h)** scPDSI. EWW, LWW, and TRW represent the earlywood width, latewood width, and total tree-ring width, respectively. The correlation analyses were conducted for the period 1953–2005 for the scPDSI, and for 1957–2005 for the other climatic factors. NELR, SP67, and SPA50 indicate the three detrending

5    methods: (1) negative exponential functions combined with linear regression with negative (or zero) slope (NELR), (2) cubic smoothing splines with a 50 % frequency cutoff of 67 % of the series length (SP67), and (3) age-dependent splines with an initial stiffness of 50 years (SPA50). STD and SSF indicate the two standardization methods "standard" and "signal-free", respectively. The correlation coefficients are reflected by the colorful and different-size circles, which can be referred to the color bar as shown at the bottom of the figure. The squares filled with light and dark gray color indicate that the correlation

10   coefficients are statistically significant at the 0.05 and 0.01 level, which are tested using the Monte Carlo method (Efron and Tibshirani, 1986; Macias-Fauria et al., 2012).

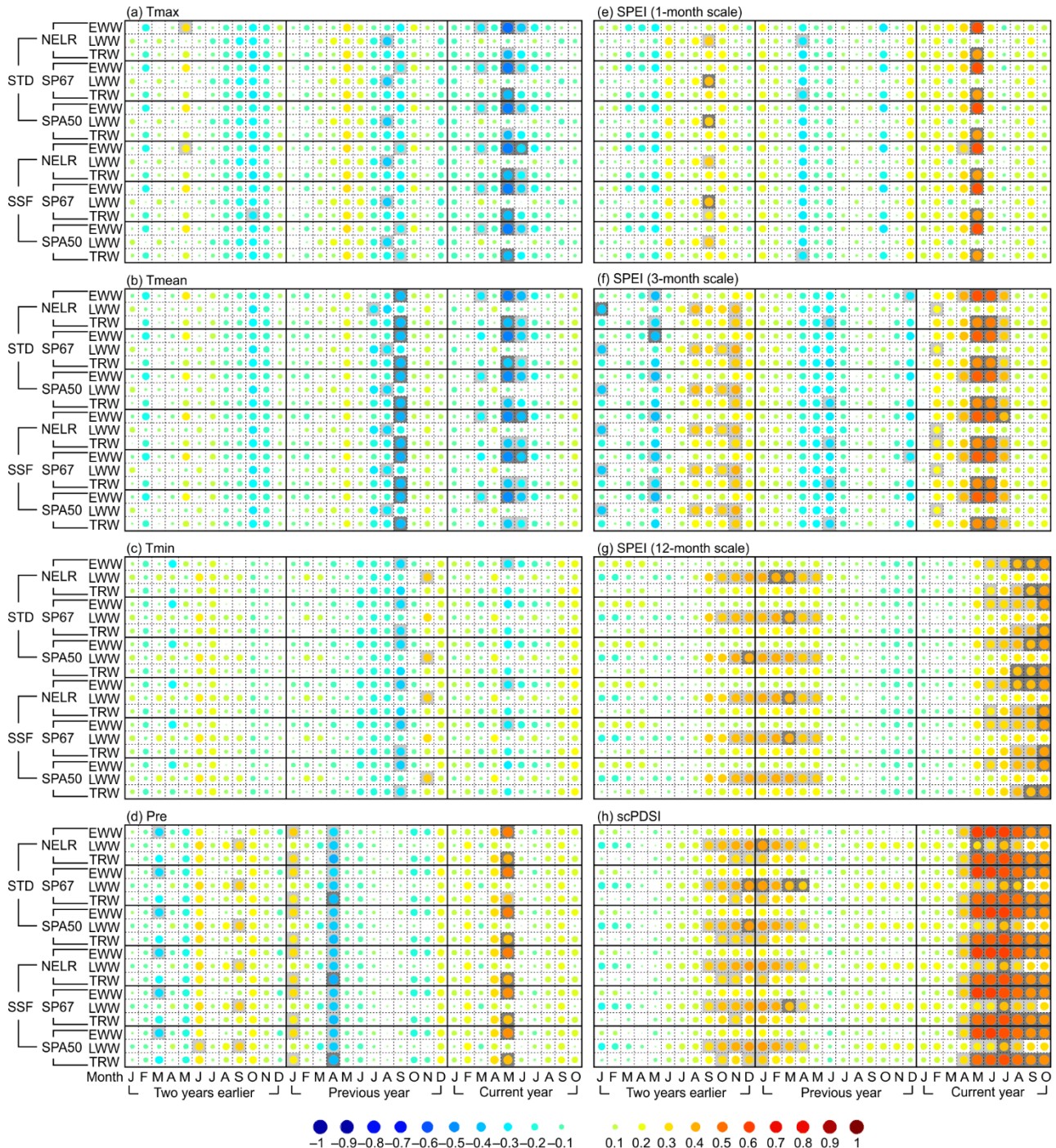

**Figure 4.** Matrix plots for the correlation coefficients between the prewhitened and linearly detrended tree-ring width chronologies and monthly climate time series from January of two years earlier to October of the current year. The climatic

factors are monthly **(a)** maximum temperature (Tmax), **(b)** mean temperature (Tmean), **(c)** minimum temperature (Tmin), **(d)** total precipitation (Pre), **(e)** SPEI of 1-month scale, **(f)** SPEI of 3-month scale, **(g)** SPEI of 12-month scale, and **(h)** scPDSI. EWW, LWW, and TRW represent the earlywood width, latewood width, and total tree-ring width, respectively. The correlation analyses were conducted for the period 1953–2005 for the scPDSI, and for 1957–2005 for the other climatic factors. NELR,

5 SP67, and SPA50 indicate the three detrending methods: (1) negative exponential functions combined with linear regression with negative (or zero) slope (NELR), (2) cubic smoothing splines with a 50 % frequency cutoff of 67 % of the series length (SP67), and (3) age-dependent splines with an initial stiffness of 50 years (SPA50). STD and SSF indicate the two standardization methods "standard" and "signal-free", respectively. The correlation coefficients are reflected by the colorful and different-size circles, which can be referred to the color bar as shown at the bottom of the figure. The squares filled with

10 light and dark gray color indicate that the correlation coefficients are statistically significant at the 0.05 and 0.01 level, which are tested using the Monte Carlo method (Efron and Tibshirani, 1986; Macias-Fauria et al., 2012).

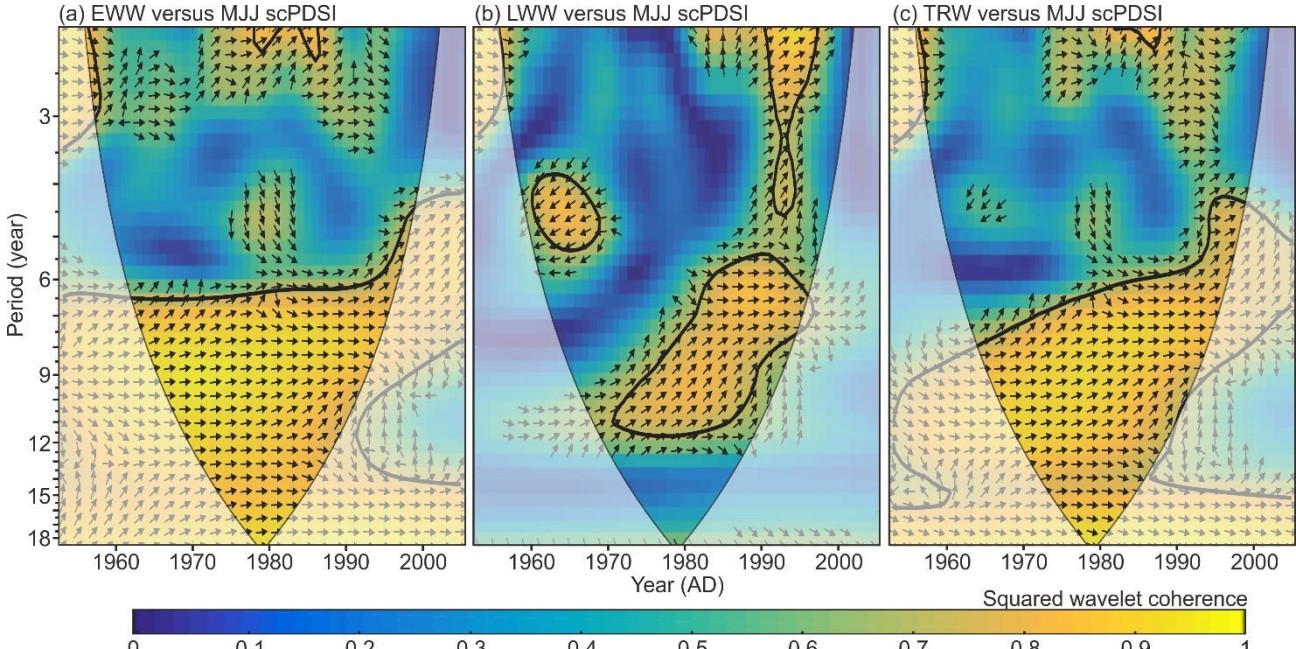

**Figure 5.** Squared wavelet coherence and phase relationship between the NELR based tree ring-width STD chronologies and MJJ scPDSI. **(a–c)** represent the results for EWW, LWW, and TRW, respectively. The color bar indicates the squared wavelet coherence. The arrows indicate the phase relationship with in-phase (anti-phase) pointing right (left), and MJJ scPDSI leading (lagging) tree-ring width with 90° pointing straight up (down). The thick contour indicates the 5 % significance level against red noise. The cone of influence (COI) where edge effects might distort the picture is marked in a lighter shade.

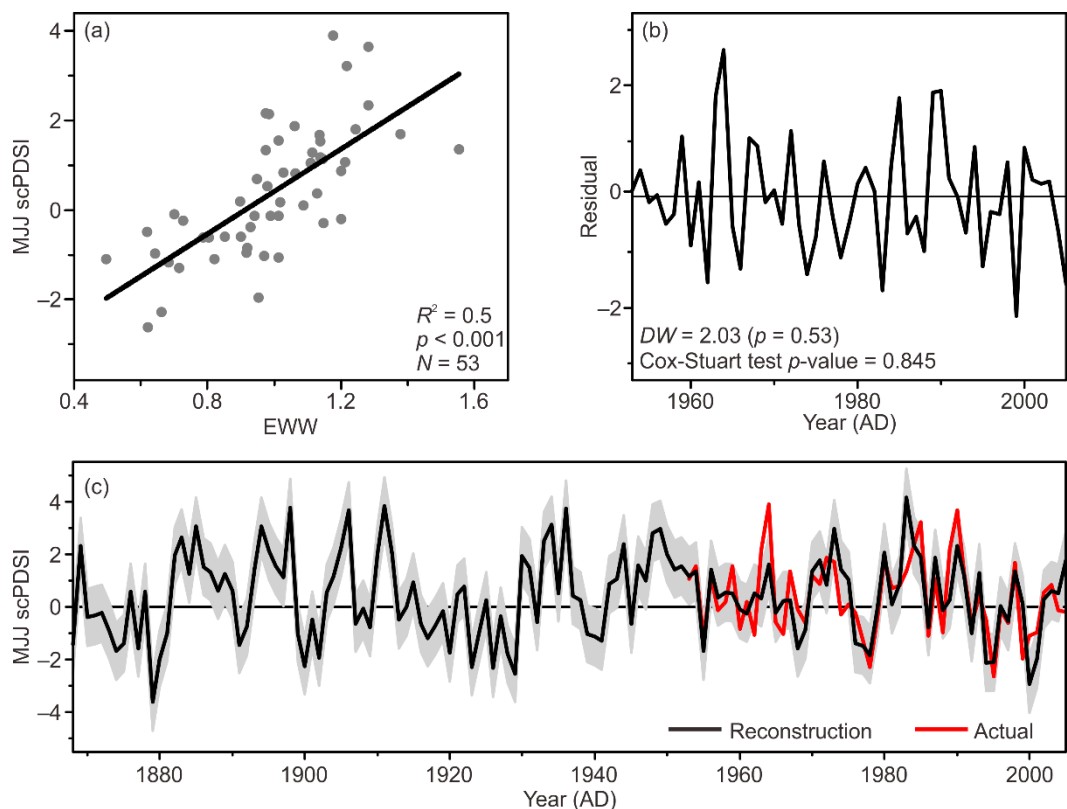

**Figure 6.** MJJ scPDSI reconstruction using NELR based EWW STD chronology. **(a)** Scatter diagram for the period 1953–2005, and **(b)** the resulting residuals. **(c)** MJJ scPDSI reconstruction (black line, after variance adjusted) and instrumental MJJ scPDSI (red line). Shaded area denotes the uncertainties of reconstruction in the form of ± 1 root mean square error.

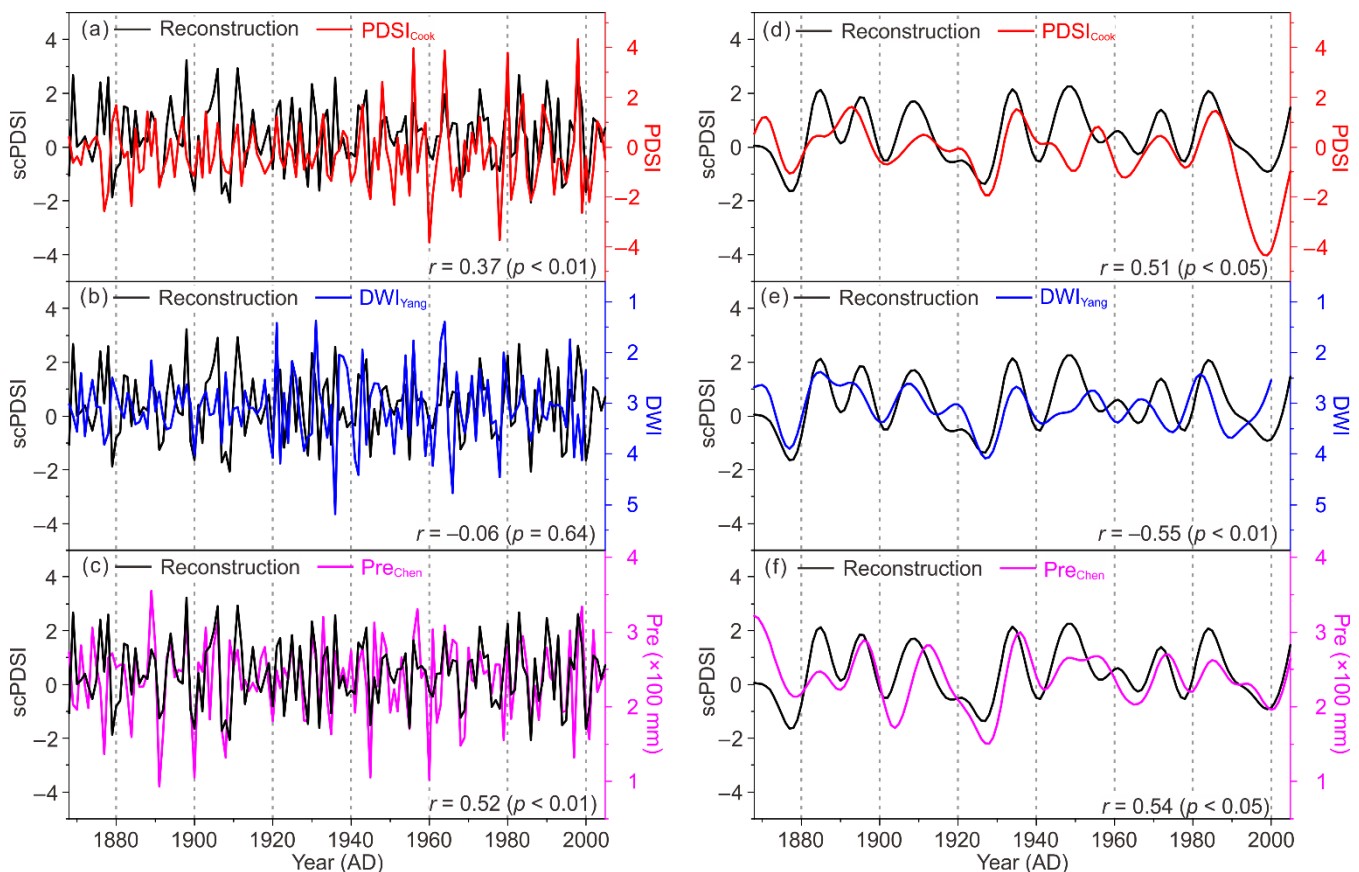

**Figure 7.** Comparison of the reconstructed MJJ scPDSI (black line) with other hydroclimatic reconstructions in adjacent regions on the interannual (left panels), and decadal and longer timescales (right panels). The referenced reconstructions are **(a, d)** June–August PDSI of MADA NO. 370 point (Cook et al., 2010), **(b, e)** reversed DWI (Yang et al., 2013), and **(c, f)** TRW based April–June precipitation (Pre) reconstruction (Chen et al., 2016b). The interannual and decadal and longer fluctuations were separated using the adaptive 10 point "Butterworth" low-pass filter with 0.1 cutoff frequency (Mann, 2008). $r$ represents the Pearson correlation coefficient between the reconstructed MJJ scPDSI and other hydroclimatic reconstruction over their common period. The significance level for all correlation coefficients were tested using the Monte Carlo method (Efron and Tibshirani, 1986; Macias-Fauria et al., 2012).

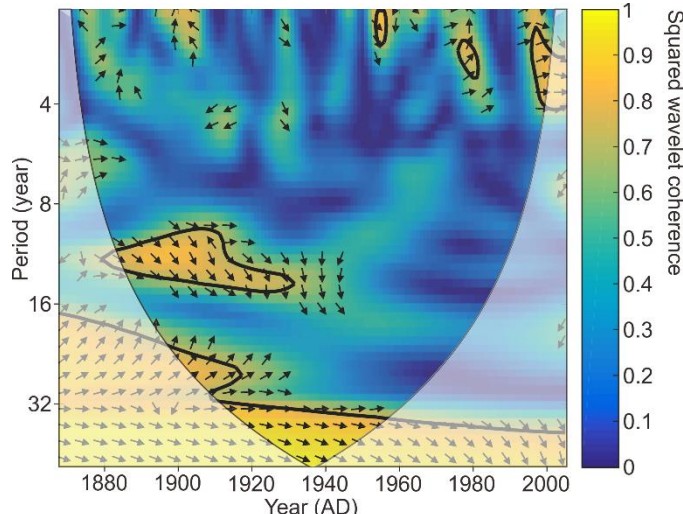

**Figure 8.** Squared wavelet coherence and phase relationship between the reconstructed MJJ scPDSI and EASMI (Zhao et al., 2015). The color bar indicates the squared wavelet coherence. The arrows indicate the phase relationship with in-phase (anti-phase) pointing right (left), and EASM leading (lagging) scPDSI with 90° pointing straight up (down). The thick contour indicates the 5 % significance level against red noise. The cone of influence (COI) where edge effects might distort the picture is marked in lighter shade.

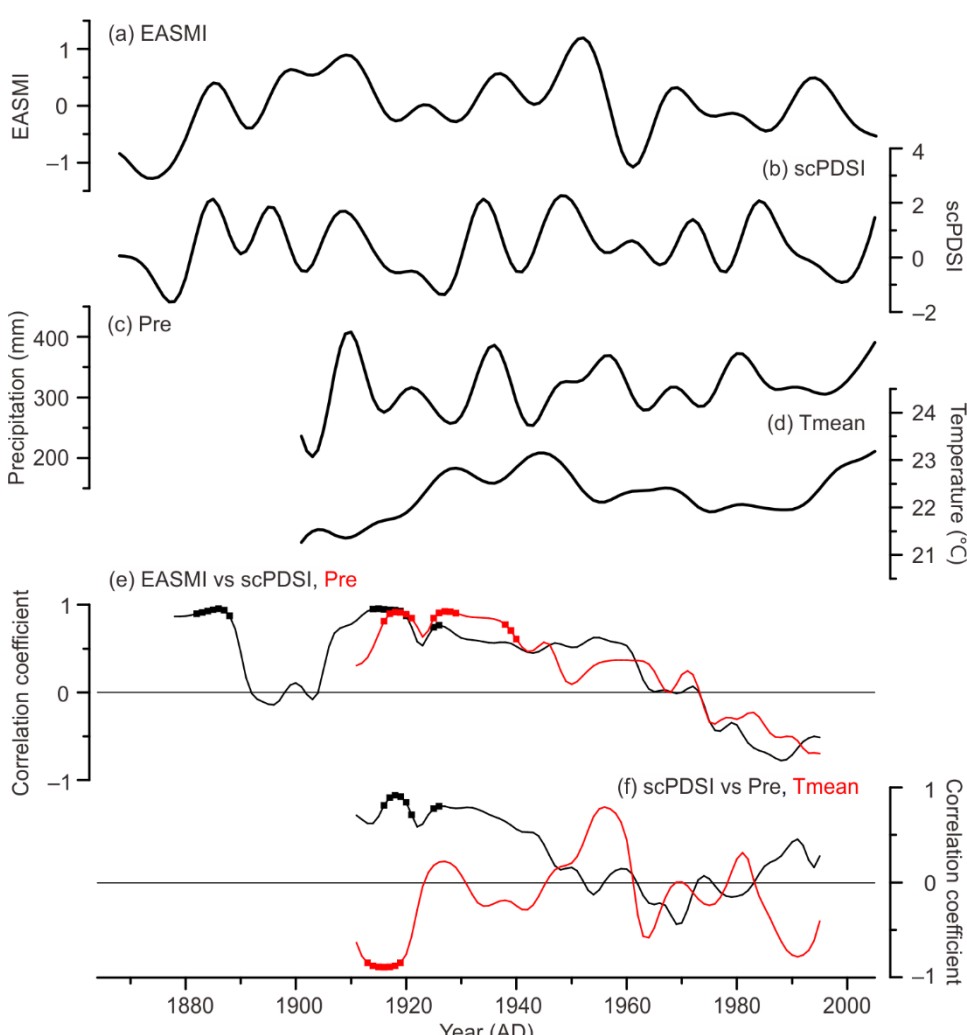

**Figure 9.** Comparison between the decadal and longer fluctuations of May–July **(a)** EASMI (Zhao et al., 2015), **(b)** reconstructed scPDSI, **(c)** precipitation (Pre; GPCC v7; Schneider et al., 2015), and **(d)** mean temperature (Tmean; CRU TS 4.01; Harris et al., 2014) over the area of reconstruction. **(e)** 21-year moving Pearson correlation coefficients of the decadal-filtered EASMI with scPDSI (black), and Pre (red). **(f)** 21-year moving Pearson correlations of the decadal filtered scPDSI with Pre (black), and Tmean (red). The decadal and longer fluctuations were derived using the adaptive 10 point "Butterworth" low-pass filter with 0.1 cutoff frequency (Mann, 2008). Statistically significant ($p < 0.05$) correlations are denoted as squares, which were tested using the Monte Carlo method (Efron and Tibshirani, 1986; Macias-Fauria et al., 2012).

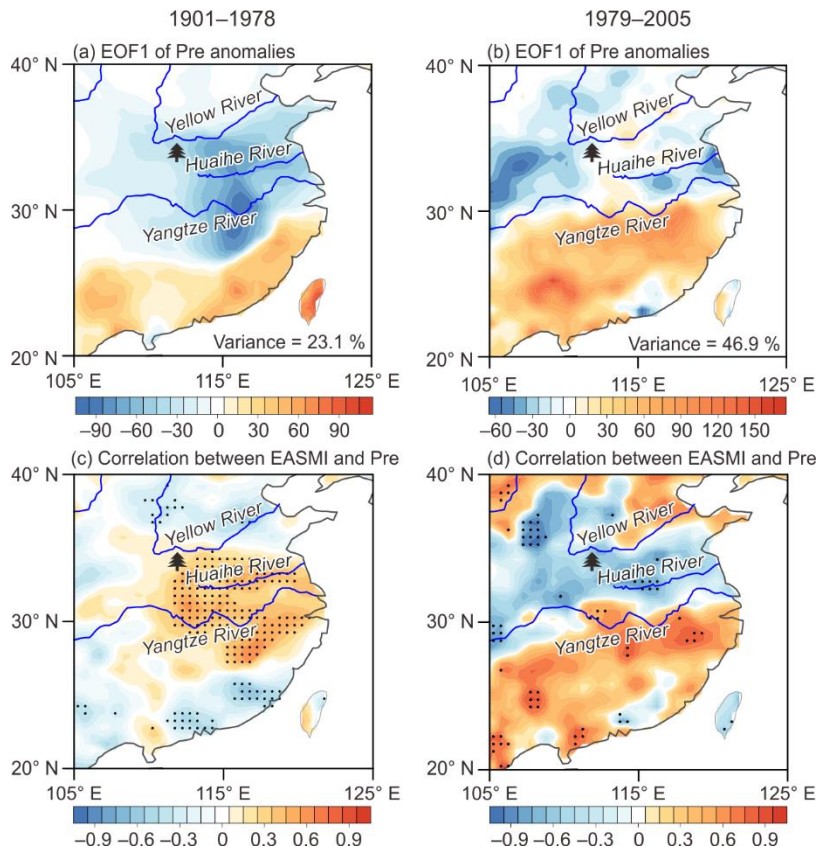

**Figure 10. (a–b)** The leading empirical orthogonal function (EOF) modes of decadal filtered May–July GPCC Precipitation anomalies for the periods 1901–1978 (left panel) and 1979–2005 (right panel). The color bar indicates the EOF values. **(c–d)** Spatial correlations between the decadal filtered May–July EASMI defined by Zhao et al. (2015) and precipitation (Pre) for the periods 1901–1978 (left panel) and 1979–2005 (right panel). The color bar indicates the correlation coefficient, dots indicate correlation significant at $p < 0.1$. Decadal-scale and long-term fluctuations of precipitation were derived using the adaptive 10 point "Butterworth" low-pass filter with 0.1 cutoff frequency (Mann, 2008). The tree symbol denotes the study region.