# Peer review of "Early summer hydroclimatic signals were well captured by tree-ring earlywood width in the eastern Qinling Mountains (central China)"

_Climate of the Past, 2018_

## Referee Comment (RC1) · Anonymous Referee #1 · 23 Nov 2018

Dear editor and authors manuscript Warm-season hydroclimate variability in Central China since 1866 AD and its relations with the East Asian Summer Monsoon: evidence from tree-ring earlywood width

Thank you for the task of reviewing the manuscript "Warm-season hydroclimate variability in Central China since 1866 AD and its relations with the East Asian Summer Monsoon: evidence from tree-ring earlywood width". The report is interesting and attempts to provide new exiting information of the application of traditional proxy-parameters derived from tree rings and at the same time attempts to provide information on the relationship between hydroclimate and the Eastern Asian Summer Monsoon (EASM). PDSI was used before in relationship to EASM at a broader scale by Cook et al. (2013), Deng et al. (2013), been applied before The manuscript is very interesting, tidy presented, with interesting figures. The work is ambitious and reach partly the objectives. I consider that the methods are appropriate to a great extent but not determinant to fully accept the conclusions of the study. The main problem as I see, is that the authors attempted to do two papers in one, one on the quality of the signal detected by different tree ring parameters, and one on the relationship of the reconstructed regional reconstruction. These are well reflected in the objectives. As a consequence, each aim is partially achieved, but not beyond doubts.

1- For the aim n1, (1) " (To) compare the climate sensitivity of tree-ring parameters earlywood width (EWW), latewood width (LWW), and total tree-ring width (TRW) in P. tabulaeformis at BYS and LCM" (where BYS and LCM two study sites). The authors compare tree ring data with means of temperature, precipitation totals and hydroclimatic index scPDSI. This aim is partially reached by the authors. It needs to be completed with further assessment of LWW and TRW parameters have significance, but are left aside for the more sensitive EWW and not further analysed. The probable relationships at different frequencies (interannual, to decadal) are tested only very succinctly with no exploration on the possible lags.

Moreover, only one detrending procedure was reported, a rather conservative one, not that it is wrong, but certainly other routines should tried when investigating aim 1. In this case, the frequency responses of each of the parameters tested should have been analysed and tailor-made detrending options to preserve best the signal characteristics The climate data should also be enhanced, different temperature patterns to start, min-max temperature and different precipitation indices.

Since there is no mention of the detrended interannual correlations except as in figure 8b (this is not mentioned in the methods) or the lower frequencies the exploration of this frequency domain can be seen as incomplete. Please see through to discuss the

differences of why PDSI indices are of higher relevance than precipitation alone mostly if tree rings series are irresponsive to precipitation. It is still unclear whether the partial correlation tests were run for precipitation and temperature excluding PDSI, etc please explain.

The authors indicate that MJJ (early season moisture availability) can be driver for the growing season increment of EWW. It can be considered that previous years moisture also affects the present year increment (see Fritts 1976) for example. The correlations tested start from July in the previous year. This means that April -Maj and June one or two years before can have importance. If this analysis is done please present the results. If it is not done yet please add it to the report. Regarding this problem, I may suggest the authors do additional tests either wavelet analysis, or evolutionary and moving intervals as those available in Dendroclim package (Biondi and Waikul, 2004) on longer temporal extension data. On the other hand, the positive correlation of LWW with PDSI indicates that there is an effect of this index on tree growth at some point in the growing season. The relationship between August temperatures on LWW with the previous year may be at least discussed. The opposite patterns of correlations found for precipitation and temperature in May (current growing season) indicates that trade-off mechanisms between these two factors and photosynthesis are in action through the beginning of the growing season. This may perhaps be clarified with extending the study period to two years before the growing season as well as testing residual chronologies against residuals of the climate data

2- " (To) Attempt to reconstruct regional hydroclimate variability using the parameter that contains the strongest hydroclimate signals". I think this is what the authors really had in mind when writing the report. I think it is brave to attempt to reconstruct regional features based on two sampling plots (33 trees) merged, located in the edge of the region in focus. Let alone to call it regional or local, to reach wider spatial representation more proxy data should be added. And previous to merge these datasets, more tests could have been attempted to see if both sites have same climatic signals. This comment is grounded on the small sample size, its only 33 individuals that can be deeply explored.

3- "To explore the relationship between reconstructed scPDSI with EASM", I understand the need to use EASM. This exploration is also succinct. But, it can and should be explored more in detail. With that in mind, almost trivial analysis are well tested and available: e.g. evolutionary response, moving intervals, coherency and wavelet analysis among others. The aim is to find synchrony (asynchrony) between datasets and extreme episodes that can be used to link two signals. These tests can really help to clarify when and how these signals could have been related and the stability of the relationship. To achieve this aim, I consider that other environmental signals with their lags should be ruled out as well. The authors mention other circulation patterns that are expected to influence the climate in the study area.

4- . 5-

Once these issues are solved, the authors will have material to two good papers: one on comparison between two or three tree ring parameters and one on the reconstruction of scPDSI and its subsequent comparison with the EASM and other atmospheric circulation patterns. I consider that the authors should take a decision on this issue and work on these alternatives separated. Each of these alternatives are promising contributions to the scientific community. Further, I provide detailed comments that may improve the article readability and content to rise its quality to a more publishable level.

Page 1 lines 15-18: Please be so kind to avoid redundancy. Page 1 line 16, MJJ scPDSI was used to denominate both the reconstruction and the scPDSI data targeted which made it rather confusing. Please use other denomination for the reconstructed data.

Introduction. Generally, the introduction is somewhat confusing, mostly due to alternation of subjects either focusing on hydroclimatic data or the EASM. Then the real

product of this article is a reconstruction hydroclimatic patterns, or an attempt to provide a predictor for the EASM, or comparisons between TRW, EWW and LWW. The authors claim that a comparison of the sensitivity to climate patters is the first objective, then the introduction should start in that way, and not focusing on EASM or scPDSI indices. WDI should be properly introduced and described.

Page 1, lines 24-25. Please consider explain the frequency domain of these examples as well as the temporal extension. If the aim is decadal to interdecadal variability, the authors could explain these anomalous events in this frequency context. Anomalous in terms of strength of the wind? The timing in the season? The spatial extension? Please explain. Line 29. "enabling it possible to study. . ." Please reword. Page 2, lines 4-5. The study is not focused on comparison with other proxies please reword. Page 2 lines 14-15. "and suggested the use of tree-ring stable isotopes to capture hydroclimate signals" Is this sentence relevant to the study? It suggests that the study focuses on these proxies. Page 2, line 20 . "These findings inspired us reconstructing hydroclimate variations. . ." please change to inspired us "to reconstruct. . ." please consider that reconstructions of past climate can not be achieve by inspiration alone. Intensive experimentation is a previous process in such an attempt. More over this sentence introduces the study aims, but later, the authors continue with introductory facts. Please consider to move this sentence further in the introduction.

Methods The authors are too general in the description of the methods. Please be specific to guarantee reproducibility of the results.

Page 3 lines 3-9: Please indicate the extension of the datasets do they start in 1887? Please indicate correlation values of detrended data, either resuduals or first differences, otherwise is a trend relationship that the authors are describing. Page 3 line 19-27. With aims of reconstruct climate data, would it not be better to keep two separate chronologies and use them as independent predictors to the PDSI? Provided that there are issues on the signal strength and intercorrelation between the two datasets may expected to be higher due to the distance between sites, whereas it may be expected to have different climatic signal due to the altitude difference... Page 3 lines 32 to Page 4 line 1: "and were quality checked before release" vague sentence and perhaps not really relevant as written here if the authors of this article have not done this quality check. What do the authors mean with quality check? Is the data homogenised in any way? Page 4 line 4. PDSI is not described in the introduction either its application in relevant articles in the area. Please be so kind to complete or specify. Page 4 lines 7-10. Please be specific what frequency domain is tested in the correlation test. Only data with no autocorrelation can give interannual responses without low frequency noise. If the standard versions of the chronologies were used the authors, they should indicate the possibility of inflated correlation values due to the slope effects of the curves. Page 4 lines 8-10. Please be more specific on what limiting factors, since the authors are performing the analysis at this stage, do they assume hydrological deficit is a limiting factor? Or temperature alone? Regional means, or extremes, etc.

Page 4 line 11: "For hydroclimate reconstruction", grammatically incorrect, please revise. Page 4 line 13: Please specify the periods which were used to split the data. Page 4 line 13: The authors could be so kind to add Durbin Watson test and Cox and Stuart Tests for the autocorrelations of the regression residuals. Page 4 Lines 13-14. Please be specific: What spatial data were compared the reconstructed time series with? Page 4 line 16. Please explain the criteria for selection of the spatial extension of the scPDSI data used in the study. Page 4 lines 21-24. This description introduces the reader to EASMI indices and should be properly described in the introduction. If you please. Page 4 lines 15-20. Could you please indicate the length of the time series named here. Page 4 line 21. Vague sentence since the term "notions" is confusing in this context, please reword. Page 4 lines 24-25. Please describe how this index was calculated, even if it is described in Zhao et al. (2015) Page 4 line 25. "the used 200 ha..." Please remove "The used" Page 4 line 31. First differences or trends? Please see comment on this issue above reference to the page 4 lines 7-10. Page 4 line 30-34. I Don't understand, is this only one procedure? correlation tests on FFT filtered series? And that is why the authors adjusted the degrees of freedom, right? Page

4 Line 34 to page 5 line 5: Is this the spatial correlation the authors used Climate explorer suite? Please explain how this was done, for example, lags, filtering, first differences, etc. Page 4 line 34. These datasets can be used to represent temperature and precipitation, rather than "reproduce". On the methods section, the descriptions of the data are good, but can be favourable to present it as well in a table. In addition, pleas be so kind to check for repetition in lines 4-6 and 16-17 in page 4. Page 5 lines 8-9: If the extension of the chronology are not specific results of this reaserch should be stated in the methods section. Moreover, these descriptions temporal extension of the data, EPS, Rbar, mean, etc are better presented in a table. If the authors will keep the paragraph, please add some values, these give base to the comparison between chronologies. For example, how much stronger were the common signals of EWW? Page 5 line 20: Please revise the grammar, "time stable" change to "is more stable through time than..." but then what do the authors mean with this? Could you please prove this with values. Page 5 line 23. "By contrast, LWW almost has no significant correlations" please add the values to make it comparable, and change "almost has no" to "has almost no..." Page 5 line 24. A conceptual observation, LWW can not induce anything... The researchers included LWW information in TRW information. The effect is understandably a decrease of climate sensitivity for the months and frequency tested. But please consider to test the data with no trends.

One more observation: the positive correlations between tree ring data and temperature and PDSI in months other than growing season can be seen as an alarm. Is it possible that there is an artefact rising the correlation values? Following the same reasoning, the spread of the correlation values is quite low both before and after the growing season, and I do not think that the trees continue photosynthesizing in December? This issues needs to be explored and analysed more deeply before publication.

Page 6 line 5, did the authors consider two chronologies for predictors of scPDSI?

Page 6 lines 12-13. "We restored the variance of reconstruction... " Do the authors mean scaled? Also consider pleas to add "the" before "reconstruction". Page 6 lines

15-16. Please indicate the frequency domain the correlation is tested on. Page 6 lines 19-21. Please consider the number of datasets used in Cook et al, 2010 (>300) in relation to this study where the authors used two chronologies, it could be argued that the spatiotemporal signal strength in this study is restricted to the area shown in the figure 1. But also, be so kind to consider the different target seasons of these datasets (MJJ and JJA). In relation to the figure 1(b): Do the authors refer to NADA dataset only to the grid point indicated in the map with the red triangle? If so, it is not clear in the text, or in the figure. I also consider that a suggestion that NADA is biased and the results presented here are more correct is premature (just on regard of the sample size). Page 6 lines 21-22. As mentioned before, please report the frequency domain of the test. Page 6 lines 25-26. Please notice that Van der Schrier et al. (2013) explains values between 2 and 3 (-2- -3) as moderated wet (moderately dry). Since the authors are using their data is worth to be consistent with their definition. Page 6 lines 26-32. It could be valuable if the authors could show some statistics (significance) of these coincident events and if possible, described events shown in different sources that are not detected by the reconstruction. Page 7 lines 5-6. Very interesting! Please consider explain in the methods how this breaking point (1956) was established. Page 7 lines 12-13. Please explain this claim, what is the importance of a dipole pattern? Is it miningful? Is a dipole pattern contrasting to conditions previous 1950s decade? Page 7 line 13. Please demonstrate thi claim with some tests. Page 7 line 14. Please change "and there are no significant spatial pattern changes" for "and there are no significant changes on the spatial patterns" Page 7 lines

Figures Figure 1(a) units or information on the colour bar are missing. Figure 1(b) Please add code or name of the stations, altitude can be also relevant. Since the EASM is relevant to the article can be good to indicate the spatial influence of this phenomena in the map.

Figure 1 caption Page 17 line 3. Please change "Cycle" for "circles". "Monsoon atlas. . ." ". . .grid point triangle" please reword this sentence, since it is not altogether clear what

the authors mean. The last sentence "and the range ..." please clarify that is a selection taken from Van der Schrier et al., (2013) larger dataset.

Figure 2. Please list the stations if possible, with the temporal extension. Figure 3. Please change "piece" for "section". These examples usually list the sample ID. Figure 4. Figure caption line 4. Please change "size" for "depth". Figure 5. Is this figure really relevant? Please write the names of the datasets. Figure 6. This is a key figure for the study. It must be complete. Please add at least from April in the previous growing season, and I wish to suggest the authors to add 2 years before the current growth year. I assume these correlations are run with the standard chronologies which probably contain a significant amount of trends. A figure similar to this could be added with prewhitened tree ring and station data for each chronology. Figure 7. It is a very interesting figure. This can be completed with LWW and TRW information, to rule out the possibility that there has been loss of signal for LWW. The figure itself is good and illustrative but highlights that no lags were tested.

Consider that this figure is made on time span after 1956, a date that the authors claim there is a change in the relationship of hydroclimatic variability and EASM. Thus this figure is restricted to "actual" conditions and not useful to illustrate past relationships. This is a subtle problem that challenges the temporal stability of the relationship between the datasets tested (tree-rings and climate).

Figure 8. Caption: a, "Raw time series" is this the raw chronology? or standard chronology "raw"?. b, 1st order difference over raw time series or standard chronologies? Please add some statistics on the figure, and analysis of residuals. Figure 9. It is demanding for the reader to guess all the time whether is scPDSI reconstruction or original data. Please make a denomination of the reconstructed index. Please add the authors in the corresponding axis of the charts. There is a lag between precipitation and scPDSI (d). This is not discussed at all, as the information in this figure is hardly integrated in the manuscript. Figure 10, caption. Please indicate what type of filter. Pearson correlation? Figure 11, please indicate what represent the colour bar in the

figure. Caption: Please indicate what type of filter, reconstructed scPDSI?, source of the datasets: "author et al (year)", Mean (?) temperature. This figure answers the question "is any change of atmospheric regime in the EASM area" not altogether relevant with the objectives, since the time domain is marginal within the reconstruction period. Are the rivers set as geographical reference?

Missing in the manuscript:

The more urgent motivation, local reconstruction, should have more local facts. More accurate description of the data, what was originally used for and why was it relevant to this one study. Often reports do not include such information, but since the dataset is small, it is worth to convince the reader of the robustness of the data. An overview table and figures with the chronology information: this because different tree ring datasets are compared. This comparison must be done in deep. An overview table with the climate data used An overview of the data used for comparison (discussion) Better descriptions of the methods used (more accurate) Better descriptions of some of the datasets e.g. DWI. The relevance of the findings. Why are these results valuable? Please be so kind to explain. Axis information in the figures (colour bars information) Text information within the figures. Acronyms to the specific datasets, two datasets can not be called in the same way. Please fix this detail.

---

## Short Comment (SC1) · 1 Dec 2018

The authors selected the earlywood width to reconstruct the early summer drought in eastern Qinling Moiuntain. Their reconstruction follows standard methods and presented new datasets. The authors also have some discussion on the drought regimes in relation to easm. I agree with publication after a revision. I have some suggestions as shown below.

1. Line 35, p2, I feel that there are many tree-ring data in southern China is related to

hydroclimate. This is not rare.

2 line 21-24, p4, you mentioned the monsoon indices by Wang and then you actually used the one by Zhao. I think you just need to mention the one by Zhao. You may need to introduce why this one is better and then you select it, but not just because it is longer. You may also do not need to detail the reanalysis data that used to derive the indices. I suggest you to focus on the introduction of the key part of this index.

3 line18-19, I do not understand well on why you calculate partial correlation with temperature and precipitation, because actually pdsi are calculated based on the temperature and precipitation. So they are related. Please add some explanations.

4 is it common for earlywood to respond to early summer moisture but the latewood has no response? It would be interesting to add more interpretation on this part in the revision. I am curious why the latewood has no correlation at all.

5. I feel that figure 3 can be moved into the appendix.

6. it is interesting you found a shift in correlations in the 1950s. This may be related to a shift between monsoon and local precipitation. You can use long instrumental precipitation to test their relationships. It is also helpful to add more discussion in relation the dipole pattern. This can be a novel point of the study

7. The authors identified 10 anomalously dry years and 11 anomalously wet years in the reconstruction period, and most of the anomalously dry (wet) years could be verified by corresponding descriptions in historical documents. Seems that there are some mismatches with the reconstruction, such as the flooding in 1954 and 1998 and the drought in 1958. Please add more discussion.

---

## Referee Comment (RC2) · Anonymous Referee #2 · 29 Dec 2018

Using reanalyzing tree-ring material from Shi et al. (2012), this manuscript found that EWW was a better hydroclimatic index in central China than TRW and LWW. The author reconstructed the growing season scPDSI based on standard procedure of dendroclimatology, and proved the fidelity of the reconstruction. I totally agreed another reviewer's comments, and suggest publication after they fully consider the comments. My confusions are listed as following: 1. Pinus tabulaeformis may stop radial growth during November and December in the study area. It may be unreasonable to consider these months for pearson's correlation in line 8 of page 4 and Figure 6. 2. The MJJ

scPDSI was reconstructed based on downloaded scPDSI (32°-35°N, 110°-112°E), it is unnecessary to compare them again in Figure 9. r value in the contents is enough. 3. I'm confused about the contents in lines 18-19 of page 5. Since the calculation of scPDSI was based on multi-proxies including precipitation and temperature, the result of partial correlation (r = 0.59, p < 0.01) that removed the effects of temperature and precipitation could only indicate that factors other than precipitation and temperature control tree-ring growth. It's not helpful for your conclusion. 4. After you reconstructed MJJ scPDSI using the linear model, do you deal the reconstruction with special method to make it match the variance of instrumental scPDSI? and how? (Page 6, line 12-13). I'm interested in it. 5. The reasons for the unstable relationship between scPDSI reconstruction and EASM are simply discussed. Is it caused by the calculation method of EASM? Because there are several EASM indices calculated in different ways. Do you try to compare your reconstruction with other EASMI, such as EASMI from Jianping Li?

---

## Author Comment (AC1) · 11 Mar 2019

Response to referee comment 1

1. Referee's comment: Dear editor and authors. Thank you for the task of reviewing the manuscript "Warm-season hydroclimate variability in Central China since 1866 AD and its relations with the East Asian Summer Monsoon: evidence from tree-ring earlywood width". The report is interesting and attempts to provide new exiting information of the application of traditional proxy parameters derived from tree rings and at the same time

attempts to provide information on the relationship between hydroclimate and the Eastern Asian Summer Monsoon (EASM). PDSI was used before in relationship to EASM at a broader scale by Cook et al. (2013), Deng et al. (2013), been applied before. The manuscript is very interesting, tidy presented, with interesting figures. The work is ambitious and reach partly the objectives. I consider that the methods are appropriate to a great extent but not determinant to fully accept the conclusions of the study. The main problem as I see, is that the authors attempted to do two papers in one, one on the quality of the signal detected by different tree ring parameters, and one on the relationship of the reconstructed regional reconstruction. These are well reflected in the objectives. As a consequence, each aim is partially achieved, but not beyond doubts. Author's Response: Thank you very much for your comments. We have strengthened the analysis for each aim, and hope you find this revision satisfactory.

2. Referee's comment: 1. For the aim n1, (1) "(To) compare the climate sensitivity of tree-ring parameters earlywood width (EWW), latewood width (LWW), and total tree-ring width (TRW) in P. tabulaeformis at BYS and LCM" (where BYS and LCM two study sites). The authors compare tree ring data with means of temperature, precipitation totals and hydroclimatic index scPDSI. This aim is partially reached by the authors. It needs to be completed with further assessment of LWW and TRW parameters have significance, but are left aside for the more sensitive EWW and not further analyzed. The probable relationships at different frequencies (interannual, to decadal) are tested only very succinctly with no exploration on the possible lags. Author's Response: Thank you very much for suggestion. We enhanced the analysis to verify that EWW can provide much stronger hydroclimatic signals than TRW and LWW from the aspects of different frequency domain, lags and leads using the wavelet coherence method. Please refer to Line 10-12 of Page 6, Line 26-33 of Page 8, and Fig. 5 in the revision.

3. Referee's comment: Moreover, only one detrending procedure was reported, a rather conservative one, not that it is wrong, but certainly other routines should be

tried when investigating aim 1. In this case, the frequency responses of each of the parameters tested should have been analyzed and tailor-made detrending options to preserve best the signal characteristics. The climate data should also be enhanced, different temperature patterns to start, min-max temperature and different precipitation indices. Author's Response: Thank you very much for suggestion. We used other two detrending methods and signal-free method to create six kinds of chronologies for comparison, and to find out the best detrending and standardization method. Please refer to Section 2.3, Section 3.1 and Figs. 3-4 in the revision. We added the maximum temperature, minimum temperature, and the SPEI of 1-month, 3-month and 12-month to enhance the climate data. Please refer to Section 2.4, Section 3.1 and Figs. 3-4 in the revision.

4. Referee's comment: Since there is no mention of the detrended interannual correlations except as in figure 8b (this is not mentioned in the methods) or the lower frequencies, the exploration of this frequency domain can be seen as incomplete. Please see through to discuss the differences of why PDSI indices are of higher relevance than precipitation alone mostly if tree rings series are irresponsive to precipitation. It is still unclear whether the partial correlation tests were run for precipitation and temperature excluding PDSI, etc please explain. Author's Response: Thank you very much for pointing out these issues. we also calculated the correlation coefficients between the prewhitened and linearly detrended chronologies and climate data to indicate that no inflation of correlation due to the autocorrelations and trends. Please refer to Line 5-8 of Page 6, Section 3.1 and Fig. 4. Test on the lower frequencies was done using wavelet coherence method, please refer to the answer for Comment 1. In fact, the May precipiation has significant impact on tree-growth (Figs. 3-4). The reasons why EWW is still restricted by PDSI but not precipitation during June-July can be referred to Line 26-29 of Page 7 in revision. Just as the comments of RC2 and SC1, the partial correlation tests for tree-ring width and precipiation, temperature, and PDSI is unreasonable, since the PDSI is calculated based on precipiation and temperature. Therefore, we removed the partial correlation analysis.

5. Referee's comment: The authors indicate that MJJ (early season moisture availability) can be driver for the growing season increment of EWW. It can be considered that previous years moisture also affects the present year increment (see Fritts 1976) for example. The correlations tested start from July in the previous year. This means that April, May and June one or two years before can have importance. If this analysis is done please present the results. If it is not done yet please add it to the report. Regarding this problem, I may suggest the authors do additional tests either wavelet analysis, or evolutionary and moving intervals as those available in Dendroclim package (Biondi and Waikul, 2004) on longer temporal extension data. On the other hand, the positive correlation of LWW with PDSI indicates that there is an effect of this index on tree growth at some point in the growing season. The relationship between August temperatures on LWW with the previous year may be at least discussed. The opposite patterns of correlations found for precipitation and temperature in May (current growing season) indicates that trade off mechanisms between these two factors and photosynthesis are in action through the beginning of the growing season. This may perhaps be clarified with extending the study period to two years before the growing season as well as testing residual chronologies against residuals of the climate data. Author's Response: Thank you very much for pointing out these aspects needed to be considered. We extended the time period to the January of two years earlier. Please refer to Line 2-3 of Page 6, Line 10-17 of Page 8 and Figs. 3-4. We used wavelet coherence method to study the temporal stability. Please refer to the response for Comment 2. We discussed the possible reasons for the significant correlation between LWW and last August temperature, please refer to Line 10-17 of Page 8. Test on the residual chronologies and residuals of climate data was done. Please refer to the response to Comment 4.

6. Referee's comment: "(To) Attempt to reconstruct regional hydroclimate variability using the parameter that contains the strongest hydroclimate signals". I think this is what the authors really had in mind when writing the report. I think it is brave to attempt to reconstruct regional features based on two sampling plots (33 trees) merged, located

in the edge of the region in focus. Let alone to call it regional or local, to reach wider spatial representation more proxy data should be added. And previous to merge these datasets, more tests could have been attempted to see if both sites have same climatic signals. This comment is grounded on the small sample size, its only 33 individuals that can be deeply explored. Authors' Response: Thank you for pointing out the problems. In the revision, we firstly calculated the correlations between the chronologies of two sites. We found that the chronologies showed very high correlation, indicating they shared similar climatic signals. Therefore, we merged the tree-ring samples from the two sites to create a composite chronology. This can be referred to Line 20-23 of Page 4, and Table S1 in the Supplementary material. The reason for that our tree-ring sites located in the edge of focus may be because the meteorological stations utilized by CRU scPDSI dataset were unevenly distributed and mainly concentrated in the west side of our tree-ring sites. Please refer to Fig. S5 and Table S4 in the Supplementary material. To capture a regional scPDSI variation, we admitted that the sample depth is too small. In the revision, we selected the scPDSI over a smaller space for calibration. We would take more samples in the future to capture a regional scPDSI variation.

7. Referee's comment: "To explore the relationship between reconstructed scPDSI with EASM". I understand the need to use EASM. This exploration is also succinct. But, it can and should be explored more in detail. With that in mind, almost trivial analysis are well tested and available: e.g. evolutionary response, moving intervals, coherency and wavelet analysis among others. The aim is to find synchrony (asynchrony) between datasets and extreme episodes that can be used to link two signals. These tests can really help to clarify when and how these signals could have been related and the stability of the relationship. To achieve this aim, I consider that other environmental signals with their lags should be ruled out as well. The authors mention other circulation patterns that are expected to influence the climate in the study area. Author's Response: Thank you for suggestion. In the revision, we tentatively explore the relationship between the reconstructed hydroclimate variability and EASM. Firstly, we used the wavelet coherence method to test the temporal stability and lags of the relationship

between EASMI and reconstructed scPDSI. A strong in-phase relationship between EASMI and the reconstructed scPDSI was found before the 1940s on the decadal and longer timescales. And, this significant in-phase relationship was further evidenced by the 21-year moving window correlation analysis on the decadal-filtered EASMI and scPDSI. We detailly explored the causes for the unstable relationship between EASMI and scPDSI using the precipiation data. We attributed the lack of correlation between EASMI and scPDSI partly to the change of leading mode of EASM precipitation. Please see section 3.4 in the revision. The influence of other circulation patterns on the climate in our sampling sites would be studied in the future.

8. Referee's comment: Once these issues are solved, the authors will have material to two good papers: one on comparison between two or three tree ring parameters and one on the reconstruction of scPDSI and its subsequent comparison with the EASM and other atmospheric circulation patterns. I consider that the authors should take a decision on this issue and work on these alternatives separated. Each of these alternatives are promising contributions to the scientific community. Further, I provide detailed comments that may improve the article readability and content to rise its quality to a more publishable level. Author's Response: Thank you very much for your evaluation and advice. In the revision, we mainly focused on revealing the climatic significance of EWW and reconstructing the MJJ scPDSI. Further comparisons with the large-scale atmospheric circulation pattern are indeed an important task, but we have limited ability to dig into this issue at this stage, given that we have only one series based on two sampling sites, and the climate forcing are very complicated. Therefore, we only conducted a tentative exploration of the relationship between the reconstructed scPDSI and EASM (the most apparent influence factor) in section 3.4, indicating that this reconstruction could provide us some new understanding of the impact of EASM on local hydroclimatic condition. Please consider whether this part is acceptable.

9. Referee's comment: Page 1 lines 15-18: Please be so kind to avoid redundancy. Author's Response: Many thanks. It was modified. Please refer to Line 20-22 of Page

1.

10. Referee's comment: Page 1 line 16, MJJ scPDSI was used to denominate both the reconstruction and the scPDSI data targeted which made it rather confusing. Please use other denomination for the reconstructed data. Author's Response: Sorry for this. In the revision, we only used the MJJ scPDSI from CRU scPDSI 3.25 dataset for reconstruction. The comparison was deleted.

11. Referee's comment: Introduction. Generally, the introduction is somewhat confusing, mostly due to alternation of subjects either focusing on hydroclimatic data or the EASM. Then the real product of this article is a reconstruction hydroclimatic patterns, or an attempt to provide a predictor for the EASM, or comparisons between TRW, EWW and LWW. The authors claim that a comparison of the sensitivity to climate patters is the first objective, then the introduction should start in that way, and not focusing on EASM or scPDSI indices. WDI should be properly introduced and described. Author's Response: Thank you for pointing out this problem. We modified the introduction thoroughly with focusing on tree-ring directly rather than EASM or scPDSI. Please see Section 1 in the revision. The "WDI" in the comment may be "DWI" as we think, it was detailly described in the Line 18-23 of Page 5, as it was only used for comparison with our reconstruction.

12. Referee's comment: Page 1, lines 24-25. Please consider explain the frequency domain of these examples as well as the temporal extension. If the aim is decadal to interdecadal variability, the authors could explain these anomalous events in this frequency context. Anomalous in terms of strength of the wind? The timing in the season? The spatial extension? Please explain. Author's Response: This part was removed. In the revision, we start the introduction from tree-ring based reconstruction, and intra-annual tree-ring width directly. The EASM is not the key part of the introduction.

13. Referee's comment: Page 2, lines 4-5. The study is not focused on comparison with other proxies please reword. Author's Response: This part was removed, as we

focused on tree-ring based reconstruction, and intra-annual tree-ring width parameters in the introduction.

14. Referee's comment: Page 2 lines 14-15. "and suggested the use of tree-ring stable isotopes to capture hydroclimate signals" Is this sentence relevant to the study? It suggests that the study focuses on these proxies. Author's Response: Many thanks. We removed this sentence.

15. Referee's comment: Page 2, line 20 . "These findings inspired us reconstructing hydroclimate variations..." please change to inspired us "to reconstruct..." please consider that reconstructions of past climate can not be achieve by inspiration alone. Intensive experimentation is a previous process in such an attempt. More over this sentence introduces the study aims, but later, the authors continue with introductory facts. Please consider to move this sentence further in the introduction. Author's Response: Many thanks. We removed this sentence and clarified our aims only in the end of the introduction.

16. Referee's comment: Methods The authors are too general in the description of the methods. Please be specific to guarantee reproducibility of the results. Author's Response: Thank you very much for this suggestion. We added more detail descriptions in the method section including the different detrending and standardization methods, correlation analysis, prewhitening and detrending methods, low-pass filtering methods and so on. Please refer to Section 2.3 and 2.5.

17. Referee's comment: Page 3 lines 3-9: Please indicate the extension of the datasets do they start in 1887? Please indicate correlation values of detrended data, either residuals or first differences, otherwise is a trend relationship that the authors are describing. Author's Response: Thank you very much for this suggestion. Extension of the datasets were indicated in Line 13 of Page 4 in the revision. Correlation were tested based on both the original standard and signal-free chronologies and their prewhitened and linearly detrended series. Please refer to Line 20-23 of Page 4 and Table S1 in the

[Figure]

Supplementary material.

18. Referee's comment: Page 3 line 19-27. With aims of reconstruct climate data, would it not be better to keep two separate chronologies and use them as independent predictors to the PDSI? Provided that there are issues on the signal strength and intercorrelation between the two datasets may expected to be higher due to the distance between sites, whereas it may be expected to have different climatic signal due to the altitude difference. Author's Response: Please refer to responses for Comment 6 and Comment 17.

19. Referee's comment: Page 3 lines 32 to Page 4 line 1: "and were quality checked before release" vague sentence and perhaps not really relevant as written here if the authors of this article have not done this quality check. What do the authors mean with quality check? Is the data homogenized in any way? Author's Response: Sorry for this. We mean the "quality check" is the homogeneity and missing values had been checked and corrected by the China Meteorological Administration before publish the data. We deleted this sentence in the revision.

20. Referee's comment: Page 4 line 4. PDSI is not described in the introduction either its application in relevant articles in the area. Please be so kind to complete or specify. Author's Response: Thank you for this suggestion. It was done. Please refer to Line 20-23 of Page 2.

21. Referee's comment: Please be specific what frequency domain is tested in the correlation test. Only data with no autocorrelation can give interannual responses without low frequency noise. If the standard versions of the chronologies were used the authors, they should indicate the possibility of inflated correlation values due to the slope effects of the curves. Author's Response: Thank you for this suggestion. We used the prewhitened and linearly detrended chronologies and climate data to calculate the correlations. Please refer to Line 4-8 of Page 6.

22. Referee's comment: Page 4 lines 8-10. Please be more specific on what limiting

factors, since the authors are performing the analysis at this stage, do they assume hydrological deficit is a limiting factor? Or temperature alone? Regional means, or extremes, etc. Author's Response: Thank you for this suggestion. It was done. Please refer to Line 13-15 of Page 6.

23. Referee's comment: Page 4 line 11: "For hydroclimate reconstruction", grammatically incorrect, please revise. Author's Response: Thank you for this suggestion. It was done.

24. Referee's comment: Page 4 line 13: Please specify the periods which were used to split the data. Author's Response: Thank you for this suggestion. It was done. Please refer to Line 16 of Page 6.

25. Referee's comment: Page 4 line 13: The authors could be so kind to add Durbin Watson test and Cox and Stuart Tests for the autocorrelations of the regression residuals. Author's Response: Thank you for this suggestion. It was done. Please refer to Line 18-23 of Page 6, Table 3, and Fig. 6b in the revision.

26. Referee's comment: Page 4 Lines 13-14. Please be specific: What spatial data were compared the reconstructed time series with? Author's Response: It's the CRU scPDSI 3.25 dataset (van der Schrier et al., 2013). It was added in the revision. Please refer to Line 29-31 of Page 6.

27. Referee's comment: Page 4 line 16. Please explain the criteria for selection of the spatial extension of the scPDSI data used in the study. Author's Response: We selected this spatial extension of the scPDSI because the scPDSI in this area has the highest correlations with our EWW, although it was in the west side of our tree-ring sites. This may be because the meteorological stations utilized by CRU scPDSI dataset were unevenly distributed and mainly concentrated in the west side of our tree-ring sites. Please refer to Line 6-9 of page 5, Fig. S5 and Table S4 in the Supplementary material.

28. Referee's comment: Page 4 lines 21-24. This description introduces the reader to EASMI indices and should be properly described in the introduction. If you please. Author's Response: Thank you very much for this suggestion. It was done. Please refer to Line 22-25 of Page 2. Besides, we detailly described this EASMI in Line 23-31 of Page 5.

29. Referee's comment: Page 4 lines 15-20. Could you please indicate the length of the time series named here. Author's Response: It was done. Please refer to Line 15-18 of page 5.

30. Referee's comment: Page 4 line 21. Vague sentence since the term "notions" is confusing in this context, please reword. Author's Response: Thanks. It was done.

31. Referee's comment: Page 4 lines 24-25. Please describe how this index was calculated, even if it is described in Zhao et al. (2015). Author's Response: Thanks. It was done. Please refer to Line 25 of Page 5.

32. Referee's comment: Page 4 line 25. "the used 200 ha..." Please remove "The used" Author's Response: Thanks. It was done.

33. Referee's comment: Page 4 line 31. First differences or trends? Please see comment on this issue above reference to the page 4 lines 7-10. Author's Response: In the revision, we tested the correlations between EASMI and our reconstruction on different frequency domain using the wavelet coherence method. When we compared the decadal filtered EASMI, scPDSI, and Precipiation, we used Pearson's correlation analysis, and the significance of correlation coefficients were tested using Monte Carlo method. The significance of correlations between tree-ring width and climate data was also tested using Monte Carlo method. Please refer to Line 14-19 of Page 7.

34. Referee's comment: Page 4 line 30-34. I Don't understand, is this only one procedure? correlation tests on FFT filtered series? And that is why the authors adjusted the degrees of freedom, right? Author's Response: Sorry for this. Since the time series

were lowpass filtered by FFT, their degrees of freedom were changed. We test the significance of correlations between the filtered series according to Yan et al., (2003). In the revision, we tested the significance for all correlations using Monte Carlo method. Please refer to Line 14-19 of Page 7.

35. Referee's comment: Page 4 Line 34 to page 5 line 5: Is this the spatial correlation the authors used Climate explorer suite? Please explain how this was done, for example, lags, filtering, first differences, etc. Author's Response: Sorry for not detail explanation. The Climate Explorer suite cannot provide correlation on the decadal filtered series. We lowpass filtered all time series, and calculated the EOF, correlations using Matlab and draw the plots using Surfer 10. Please refer to Line 12-13 of Page 7.

36. Referee's comment: Page 4 line 34. These datasets can be used to represent temperature and precipitation, rather than "reproduce". Author's Response: Many thanks. It was done. Please refer to Line 33-34 of Page 5.

37. Referee's comment: On the methods section, the descriptions of the data are good, but can be favorable to present it as well in a table. Author's Response: Many thanks. It was done. Please refer to Table 1-2.

38. Referee's comment: In addition, pleas be so kind to check for repetition in lines 4-6 and 16-17 in page 4. Author's Response: Many thanks. We only used the CRU scPDSI dataset for reconstruction, and removed the comparison.

39. Referee's comment: Page 5 lines 8-9: If the extension of the chronology are not specific results of this research should be stated in the methods section. Moreover, these descriptions temporal extension of the data, EPS, Rbar, mean, etc. are better presented in a table. If the authors will keep the paragraph, please add some values, these give base to the comparison between chronologies. For example, how much stronger were the common signals of EWW? Author's Response: Thank you for your suggestion. As this part is not our aim in the revision, we only mentioned the extension of chronology in Section 2.3. Meanwhile, the statistics of the chronologies are presented in the form of tables shown in the Supplementary material (Table S2 and Table S3.)

40. Referee's comment: Page 5 line 20: Please revise the grammar, "time stable" change to "is more stable through time than..." but then what do the authors mean with this? Could you please prove this with values? Author's Response: Sorry for this. Here we mean that the relationship of EWW and MJJ scPDSI showed more stable through time than LWW and TRW. This was done by 21-year moving correlation analysis in the original manuscript. In the revision, we used the wavelet coherence method. Please refer to Line 26-33 of Page 8, and Fig. 5.

41. Referee's comment: Page 5 line 23. "By contrast, LWW almost has no significant correlations" please add the values to make it comparable, and change "almost has no" to "has almost no..." Author's Response: Thank you for pointing out this issue. It was done. Please refer to 2-7 of Page 8.

42. Referee's comment: Page 5 line 24. A conceptual observation, LWW can not induce anything... The researchers included LWW information in TRW information. The effect is understandably a decrease of climate sensitivity for the months and frequency tested. But please consider to test the data with no trends. Author's Response: Thank you for suggestion. The sentence has been modified. Please refer to Line 8-11 of Page 9. The tests with no trend were conducted. Please refer to the response for Comment 4.

43. Referee's comment: One more observation: the positive correlations between tree ring data and temperature and PDSI in months other than growing season can be seen as an alarm. Is it possible that there is an artefact rising the correlation values? Following the same reasoning, the spread of the correlation values is quite low both before and after the growing season, and I do not think that the trees continue photosynthesizing in December? This issue needs to be explored and analyzed more deeply before publication. Author's Response: According to the previous studies relevant to the seasonal dynamics of cambial activities in P. tabulaeformis (Line 10-17 of Page 3 in the revision), the tree would could not photosynthesize in December, and the earlywood growth could terminate in the mid-July. The significant correlations between EWW and scPDSI after the growing season may be ascribed to the characteristic of scPDSI which has a strong autocorrelation with previous months. This has been clarified in Line 31-32 of Page 7 and Line 1 of Page 8. The significant correlation between EWW and temperature in November seemed caused by the low-frequency, as there is no significant correlation was found between their first-order difference (Fig. 1). Since the Referee 2 argued that the correlations analysis between tree-ring width the and climatic factors in November and December is unreasonable, we deleted the correlation analysis in the revision.

44. Referee's comment: Page 6 line 5, did the authors consider two chronologies for predictors of scPDSI? Author's Response: Thank you for pointing out this issue. The number of our tree-rings samples were limited, especially in the LCM, where only 11 trees were only obtained. We found the tree-growth at the two sites shared very similar variations, manifesting the similar climate forcing. Therefore, we merged the tree-ring samples from LCM and BYS to get a chronology and used for calibration with scPDSI. We would take more samples in the future to capture a regional scPDSI signals. Please refer to the response to Comment 6.

45. Referee's comment: Page 6 lines 12-13. "We restored the variance of reconstruction... " Do the authors mean scaled? Also consider pleas to add "the" before "reconstruction". Author's Response: Yes, it is. Please refer to the equation (1) in the revision (Line 25 of Page 6). "the" was added.

46. Referee's comment: Page 6 lines 15-16. Please indicate the frequency domain the correlation is tested on. Author's Response: In the revision, we tested the correlation between the reconstruction and other hydroclimatic series on the interannual, and decadal and longer timescales, respectively. Please refer to Section 3.3 in the revision.

47. Referee's comment: Page 6 lines 19-21. Please consider the number of datasets used in Cook et al, 2010 (>300) in relation to this study where the authors used two chronologies, it could be argued that the spatiotemporal signal strength in this study is restricted to the area shown in the figure 1. But also, be so kind to consider the different target seasons of these datasets (MJJ and JJA). In relation to the figure 1(b): Do the authors refer to NADA dataset only to the grid point indicated in the map with the red triangle? If so, it is not clear in the text, or in the figure. I also consider that a suggestion that NADA is biased and the results presented here are more correct is premature (just on regard of the sample size). Author's Response: Many thanks for pointing out these issues. The MADA grid is labeled using a red triangle in Fig. 1. Its coordinate is included in the text. Please refer to Line 15 of Page 5. In the revision, we only discussed the mismatches between our reconstruction and the MADA and the possible reasons, and removed the argument that MADA is biased in recent decades. Please refer to Section 3.3.

48. Referee's comment: Page 6 lines 21-22. As mentioned before, please report the frequency domain of the test. Author's Response: Please refer to answer for Comment 46.

49. Referee's comment: Page 6 lines 25-26. Please notice that Van der Schrier et al. (2013) explains values between 2 and 3 (-2- -3) as moderated wet (moderately dry). Since the authors are using their data is worth to be consistent with their definition. Page 6 lines 26-32. It could be valuable if the authors could show some statistics (significance) of these coincident events and if possible, described events shown in different sources that are not detected by the reconstruction. Author's Response: Many thanks for this suggestion. It was done. Please refer to Line 12-18 of Page 10 and Table 4 in the revision.

50. Referee's comment: Page 7 lines 5-6. Very interesting! Please consider explain in the methods how this breaking point (1956) was established. Author's Response: Sorry for this. The breaking point was roughly determined visually. In the revision, we

used the wavelet coherence method, and it was showed the breaking point was located around the 1940s. Please refer to Fig. 8.

51. Referee's comment: Page 7 lines 12-13. Please explain this claim, what is the importance of a dipole pattern? Is it meaningful? Is a dipole pattern contrasting to conditions previous 1950s decade? Page 7 line 13. Please demonstrate this claim with some tests. Author's Response: The dipole pattern means the contrast precipitation anomalies over south and north part of East China, this pattern receives much attention in China because it concerns the allocation of water resources. However, this issue is beyond the scope of this paper, so we did not explain it in details and only use the phrase "dipole pattern" to describe the distribution feature of precipitation. As shown in Figs. 10b, the dipole pattern was mainly occurred since the late-1970s. In contrast, the variation of precipiation anomalies before the 1970s were similar in the south and north of the Yangtze River (Figs. 10a). We attributed the unstable relationship between EASMI and scPDSI partly to the changed leading mode of EASM precipitation. Please refer to Paragraph 2 of Section 3.4.

52. Referee's comment: Page 7 line 14. Please change "and there are no significant spatial pattern changes" for "and there are no significant changes on the spatial patterns". Author's Response: Many thanks. We have modified the discussion in this part. Please refer to Section 3.4.

53. Referee's comment: Figure 1(a) units or information on the color bar are missing. Figure 1(b) Please add code or name of the stations, altitude can be also relevant. Since the EASM is relevant to the article can be good to indicate the spatial influence of this phenomena in the map. Figure 1 caption Page 17 line 3. Please change "Cycle" for "circles". "Monsoon atlas..." "...grid point triangle" please reword this sentence, since it is not altogether clear what the authors mean. The last sentence "and the range ..." please clarify that is a selection taken from Van der Schrier et al., (2013) larger dataset. Author's Response: Many thanks. It was done.

54. Referee's comment: Figure 2. Please list the stations if possible, with the temporal extension. Author's Response: Many thanks. It was done.

55. Referee's comment: Figure 3. Please change "piece" for "section". These examples usually list the sample ID. Author's Response: Many thanks. It was done. Besides, we have moved this figure to the Supplementary material. Please refer to Fig. S1.

56. Referee's comment: Figure 4. Figure caption line 4. Please change "size" for "depth". Author's Response: Many thanks. It was done. Besides, we have moved this figure to the Supplementary material. Please refer to Fig. S4.

57. Referee's comment: Figure 5. Is this figure really relevant? Please write the names of the datasets. Author's Response: Thank you for your suggestion. We deleted this figure.

58. Referee's comment: Figure 6. This is a key figure for the study. It must be complete. Please add at least from April in the previous growing season, and I wish to suggest the authors to add 2 years before the current growth year. I assume these correlations are run with the standard chronologies which probably contain a significant amount of trends. A figure similar to this could be added with prewhitened tree ring and station data for each chronology. Author's Response: Thank you for your suggestion. Please refer to the new Fig. 4 in the revision.

59. Referee's comment: Figure 7. It is a very interesting figure. This can be completed with LWW and TRW information, to rule out the possibility that there has been loss of signal for LWW. The figure itself is good and illustrative but highlights that no lags were tested. Consider that this figure is made on time span after 1956, a date that the authors claim there is a change in the relationship of hydroclimatic variability and EASM. Thus this figure is restricted to "actual" conditions and not useful to illustrate past relationships. This is a subtle problem that challenges the temporal stability of the relationship between the datasets tested (tree-rings and climate). Author's Response: Thank you for your suggestion. Here we used the Fig. 5 to replace the original Fig. 7.

The Fig. 5 can display both the temporal stability and lags of the relationship between tree-ring parameters and MJJ scPDSI. The original Figure 7 was used to tested the temporal stability of the relationship between EWW and scPDSI, precipitation, and temperature, but not EASMI. This figure only indicated that EWW had much stable relationship with MJJ scPDSI than with precipiation and temperature. In addition, the reconstructed MJJ scPDSI can be validated by other hydroclimatic reconstructions and historical document records (Please refer to the Section 3.3 in the new revision). So, we think there is no problem in the reconstruction.

60. Referee's comment: Figure 8. Caption: a, "Raw time series" is this the raw chronology? or standard chronology "raw"?. b, 1st order difference over raw time series or standard chronologies? Please add some statistics on the figure, and analysis of residuals. Author's Response: Many thanks. It was done. Please refer to Fig. 6.

61. Referee's comment: Figure 9. It is demanding for the reader to guess all the time whether is scPDSI reconstruction or original data. Please make a denomination of the reconstructed index. Please add the authors in the corresponding axis of the charts. There is a lag between precipitation and scPDSI (d). This is not discussed at all, as the information in this figure is hardly integrated in the manuscript. Author's Response: Many thanks. The comparison with CRU scPDSI dataset were removed and the data was only used for calibration and reconstruction. Authors for corresponding reconstruction were added. The lag and mismatches were detailly discussed in revision. Please refer to section 3.3 and Fig. 7.

62. Referee's comment: Figure 10, caption. Please indicate what type of filter. Pearson correlation? Author's Response: Many thanks. It was done. Please refer to Fig. 9.

63. Referee's comment: Figure 11, please indicate what represent the color bar in the figure. Caption: Please indicate what type of filter, reconstructed scPDSI? source of the datasets: "author et al (year)", Mean (?) temperature. This figure answers the question "is any change of atmospheric regime in the EASM area" not altogether relevant with

the objectives, since the time domain is marginal within the reconstruction period. Are the rivers set as geographical reference? Author's Response: Many thanks. Since we found a decreased correlation between the reconstructed scPDSI and EASMI, we want to use this figure illustrate that decreased correlation may be associated with the change of leading EASM mode. The information of color bar, filter, rivers for reference were added. Please refer to Fig. 10.

64. Referee's comment: Missing in the manuscript: The more urgent motivation, local reconstruction, should have more local facts. More accurate description of the data, what was originally used for and why was it relevant to this one study. Often reports do not include such information, but since the dataset is small, it is worth to convince the reader of the robustness of the data. An overview table and figures with the chronology information: this because different tree ring datasets are compared. This comparison must be done in deep. An overview table with the climate data used, An overview of the data used for comparison (discussion) Better descriptions of the methods used (more accurate) Better descriptions of some of the datasets e.g. DWI. The relevance of the findings. Why are these results valuable? Please be so kind to explain. Axis information in the figures (color bars information) Text information within the figures. Acronyms to the specific datasets, two datasets can not be called in the same way. Please fix this detail. Author's Response: Thank you very much for pointing out these issues. We modified the introduction to clarify our motivation. The overview table and figures of the chronology information were added. Please refer to Fig. S4, Table S2 and Table S3. The table of climate data and reconstructions used for comparison can be referred to Table 1 and Table 2. Descriptions about DWI were added in the Line 18-22 of Page 5. The findings from the comparison between our reconstruction and other hydroclimatic reconstruction were detail discussed. Please refer to Section 3.3. Information of the axis, color bars were also added. The CRU scPDSI 3.25 data was only used for calibration and reconstruction, and the comparison was deleted.

Please also note the supplement to this comment:

https://www.clim-past-discuss.net/cp-2018-141/cp-2018-141-AC1-supplement.zip

[Figure]

[Figure]

**Fig. 1.** Linear regression between the first-order difference of the NELR based EWW STD chronology and the Tmean in November of the growth year

---

## Author Comment (AC2) · 11 Mar 2019

Response to short comment 1

1. Shorth comment: The authors selected the earlywood width to reconstruct the early summer drought in eastern Qinling Mountain. Their reconstruction follows standard methods and presented new datasets. The authors also have some discussion on the drought regimes in relation to EASM. I agree with publication after a revision. I have some suggestions as shown below. Author's Response: We really appreciate your

valuable comments and suggestions. We have carefully revised the manuscript and hope you find this revision satisfactory. Please see details below.

2. Shorth comment: line 35, p2, I feel that there are many tree-ring data in southern China is related to hydroclimate. This is not rare. Author's Response: Sorry for the inaccurate description here. We made a new summary about the hydroclimatic related tree-ring data in southern China. Please refer to line 5-11of page 2 in the revision.

3. Shorth comment: line 21-24, p4, you mentioned the monsoon indices by Wang and then you actually used the one by Zhao. I think you just need to mention the one by Zhao. You may need to introduce why this one is better and then you select it, but not just because it is longer. You may also do not need to detail the reanalysis data that used to derive the indices. I suggest you to focus on the introduction of the key part of this index. Author's Response: Many thanks for this suggestion. We added more descriptions for the EASMI developed by Zhao. Please refer to line 23-31 of page 5 in the revision.

4. Shorth comment: line18-19, I do not understand well on why you calculate partial correlation with temperature and precipitation, because actually pdsi are calculated based on the temperature and precipitation. So they are related. Please add some explanations. Author's Response: Thank you very much for pointing out this question. Just as suggested by RC2, this analysis is unreasonable, and we removed it from our revision.

5. Shorth comment: is it common for earlywood to respond to early summer moisture but the latewood has no response? It would be interesting to add more interpretation on this part in the revision. I am curious why the latewood has no correlation at all. Author's Response: Thank you for this question. In fact, the latewood still had a significant response to early summer moisture (scPDSI) as displayed in Fig. 3-5 in the revision, but the response was not as strong and stable as those found in earlywood. We added the interpretation for the less sensitivity of LWW to early summer moisture. Please

refer to Line 7-8 of Page 9.

6. Shorth comment: I feel that figure 3 can be moved into the appendix Author's Response: Thank you very much for suggestion. Since we have lots of new figures and tables, we moved the figure 3 into the Supplement material. Please see Fig. S1.

7. Shorth comment: it is interesting you found a shift in correlations in the 1950s. This may be related to a shift between monsoon and local precipitation. You can use long instrumental precipitation to test their relationships. It is also helpful to add more discussion in relation the dipole pattern. This can be a novel point of the study Author's Response: Many thanks for suggestion. We used the GPCC precipiation (1901-2005) to make comparisos between the EASMI and reconstructed scPDSI. Discussion relevant to the dipole pattern and its driving mechanisms were also enhanced. Please refer to Section 3.4 in the revision.

8. Shorth comment: The authors identified 10 anomalously dry years and 11 anomalously wet years in the reconstruction period, and most of the anomalously dry (wet) years could be verified by corresponding descriptions in historical documents. Seems that there are some mismatches with the reconstruction, such as the flooding in 1954 and 1998 and the drought in 1958. Please add more discussion. Author's Response: Thank you for this suggestion. We added some discussion about the mismatches and causes in line 14-18 of page 10 in the revision.

Please also note the supplement to this comment:
https://www.clim-past-discuss.net/cp-2018-141/cp-2018-141-AC2-supplement.zip

---

## Author Comment (AC4) · 11 Mar 2019

Response to referee comment 2

1. Referee's comment: Using reanalyzing tree-ring material from Shi et al. (2012), this manuscript found that EWW was a better hydroclimatic index in central China than TRW and LWW. The author reconstructed the growing season scPDSI based on standard procedure of dendroclimatology, and proved the fidelity of the reconstruction. I totally agreed another reviewer's comments, and suggest publication after they fully

consider the comments. My confusions are listed as following: Author's Response: Thank you very much for your review and comments. We have revised the manuscript according to your suggestions. Please see details below.

2. Referee's comment: Pinus tabulaeformis may stop radial growth during November and December in the study area. It may be unreasonable to consider these months for Pearson's correlation in line 8 of page 4 and Figure 6. Author's Response: Many thanks for suggestion. We have deleted the climate response analysis for these months. Please refer to the line 3-4 of page 6 and Figs. 3-4 in the revision.

3. Referee's comment: The MJJ scPDSI was reconstructed based on downloaded scPDSI (32âŬę-35âŬęN, 110âŬę-112âŬęE), it is unnecessary to compare them again in Figure 9. Author's Response: Thanks. We have removed this comparison. Please refer to Fig. 7 in the revision.

4. Referee's comment: I'm confused about the contents in lines 18-19 of page 5. Since the calculation of scPDSI was based on multi-proxies including precipitation and temperature, the result of partial correlation ($r = 0.59$, $p < 0.01$) that removed the effects of temperature and precipitation could only indicate that factors other than precipitation and temperature control tree-ring growth. It's not helpful for your conclusion. Author's Response: Thank you very much for pointing out this unreasonable analysis. We deleted this analysis.

5. Referee's comment: After you reconstructed MJJ scPDSI using the linear model, do you deal the reconstruction with special method to make it match the variance of instrumental scPDSI? and how? (Page 6, line 12-13). I'm interested in it. Author's Response: Yes, we have adjusted the variance of reconstructed MJJ scPDSI so that it has the same variance with the actual MJJ scPDSI. The detail method was shown as the equation 1 in Page 6 Line 25 in the revised manuscript.

6. Referee's comment: The reasons for the unstable relationship between scPDSI reconstruction and EASM are simply discussed. Is it caused by the calculation method of

EASM? Because there are several EASM indices calculated in different ways. Do you try to compare your reconstruction with other EASMI, such as EASMI from Jianping Li? Author's Response: Thank you very much for this question. We added more discussion for the unstable relationship between scPDSI reconstruction and EASM. Please refer to Section 3.4. It is well known that there are many EASMI. We could get different results if we used different EASMI. However, before the analysis, it is necessary to select an EASMI which had the best ability to capture the precipiation over East Asia and with clear physical mechanisms. The EASMI of Zhao et al., (2015) has been proved to show better ability in depicting the precipiation and temperature over East Asia compared with previous indices, this can be referred to Zhao et al., (2015). As shown in Fig.1, the EASMI of Zhao et al., (2015) had significant positive correlations ($p < 0.05$) with the May-July precipitation over the south of the Yangtze River on the decadal and longer timescales during the period 1901-2005, indicating that it can capture the Meiyu precipitation which is a good indicator of EASM as suggested by Wang et al., (2008). In comparison, the EASMI of Li and Zeng (2005) had limited ability in depicting the precipitation variability.

Reference: Li, J., and Zeng, Q.: A new monsoon index, its interannual variability and relation with monsoon precipitation, Climatic and Environmental Research, 10, 351–365, 2005. Wang, B., Wu, Z., Li, J., Liu, J., Chang, C.-P., Ding, Y., and Wu, G.: How to measure the strength of the East Asian Summer Monsoon, J. Climate, 21, 4449–4463, https://doi.org/10.1175/2008jcli2183.1, 2008. Zhao, G., Huang, G., Wu, R., Tao, W., Gong, H., Qu, X., and Hu, K.: A new upper-level circulation index for the East Asian Summer Monsoon variability, J. Climate, 28, 9977–9996, https://doi.org/10.1175/jcli-d-15-0272.1, 2015.

Please also note the supplement to this comment:
https://www.clim-past-discuss.net/cp-2018-141/cp-2018-141-AC4-supplement.zip

[Figure]

[Figure]

**Fig. 1.** Pearson correlation coefficients between the decadal-filtered May-July precipitation and EASMI ((a) Zhao et al., 2015; (b) Li and Zeng (2005)) during the period 1901-2005 over East Asia.

---

## Referee Report (RR1)

**cp-2018-141**

**Submitted on 21 Oct 2018**
**Early summer hydroclimatic signals were well captured by tree-ring earlywood width in the eastern Qinling Mountains (central China)**
**Yesi Zhao, Jiangfeng Shi, Shiyuan Shi, Xiaoqi Ma, Weijie Zhang, Bowen Wang, Xuguang Sun, Huayu Lu, and Achim Bräuning**

Climate of the Past, manuscript "Early summer hydroclimatic signals were well captured by tree-ring earlywood width in the eastern Qinling Mountains (central China)"

Authors, Yesi Zhao et al.

I want to thank the editors the possibility to review again the article from Dr Yesi Zhao and colleagues and provide below comments and suggestions to improve the work presented.

The actual version of the manuscript is much improved and therefore I can recommend it for publication after correcting some minor problems indicated below.

I think that the signal obtained in EWW is rather alike to a signal of LWW. The measuring of EWW and LWW in Tsap involves some transformation of the data into individual time series. Can it be possible that the authors mixed up the conversion of the collected data?

In addition, positive correlation with PDSI indicates that dry conditions are favourable for tree growth. PDSI positive index indicates dry conditions while negative indicates wet conditions. I understand that the index uses both temperature and precipitation data and can be the signal of temperature that dominates the correlation with positive PDSI as it can be seen in figures 3 and 4. Either case, it is important the coverability between predictor and predictand, thus using tree-ring time series as predictor of PDSI, but if the signal is dominated by temperature rather than moisture, this can be the reason of the instability of the relationship of the time series indicated in the figure 8.

Just a few comments:

Line 2 page 2, reconstructions cannot fall. Please reword.

Line 7 page 2, I think in this case precipitation "are". Please check if I am correct.

Line 11 page 2, please change unable for unsuitable.

Line 14 page 2, caused restrictions of water availability through the growing season

Line 18 page 2, are the abbreviations BYS and LCM needed? Since it is only one word for each name, I think is more reader friendly the original names, and using the locality names adds charm to the research presented.

Line 20 page 2, please consider to mention SPEI index since it was added into the analysis and first mentioned in line 33.

Line 22 page 2, please consider to give a very short background of the EASMI, what does it mean in terms of hydroclimate or temperature in the study area.

Materials and methods,

Line 3 page 3. Please consider changing "study were provided by Shi et al. (2012)." By previously presented in Shi et al., 2012

Line 13 page 3: and did not complete *around* mid-Septermber. Please reword.

Line 13 page 3 "that the cambial cells of… started and ceased in… Please be so kind to indicate what started, cell division, cell growth, activity, etc.

Line 20 page 3, Due to the characteristics of the ring anatomy, Earlywood and latewood…

Line 10 page 4, please use "by means of" instead of "via" otherwise "through consecutive iterations"

Line 12, see Osborn *and colleagues*, (1997).

Line 13, EPS computed on multiple samples per tree will be inflated creating a wrong result (inflated value) please indicate if the index was calculated on single trees or multiple samples per tree.

Line 27 page 4. Please consider that the application of the principal component analysis is to reduce the amounts of variables of an analysis, rather than assess similarity between variables. Cluster analysis can be used to assess degrees of similarity between datasets.

Line 2 page 5: SPEI and PDSI, please see the comment above on presenting both indices in the introduction.

Line 5 page 5, please reword sentence "…and calculate the duration factors…" I do not understand what the authors mean.

Line 13 page 5, please be so kind to reword: three times the word "reconstruction" in one sentence makes it difficult to follow.

Line 26 page 5, please remove "this" before EASMI and "is" after it.

Line 29 page 5. Please change "restored" for "adjusted"

Line 10 page 7. Please change "manifest" for "assess"

Line 13 page 7, please provide proper citation and names for software Matlab RM and Surfer 10.

Line 24 page 7. The positive correlation with PDSI indicates dry conditions as favourable for tree growth. Do I get that right? This means that conditions in the study area are rather too wet, and a moderate drought will be good the tree growth. Is this reasonable for an area of thin soils and well drained?

Line 15 page 8, it says "from September of two years earlier to May of last May". Please reword.

Line 5-6 page 9, please reword "Increased water deficiency due to the rising temperature and inadequate rainfall in the 5 early growing season could induce soil water deficiency thus suppressing cell expansion and cell growth in the cambium"

Line 22, page 9. Restored or adjusted?

Line 23 page 9, please add "the" before reconstruction.

Line 6 page 10. Please change "The possible reasons might be that (1) the seasons that the three reconstructions aimed at are different from ours…" for "The possible reasons might be that (1) the reconstructions aim to different target season"
Line 33 page 10: How stable in time is the spatial pattern of the EASM?
Line 25 page 11, please reword and eliminate "etc,"

Figures:
Figure 1, Southern Asia summer monsoon is not mentioned in the text
Figure 2 figure text: please add "s" to denote.

Figure 6 add "d" to shade. areas

Figure 9, figure text, citation Macias-Faura et al.","2012) please check for the right format.

Figure 10, text in the figure: Huaie River and Yangtze River in the lower left plate is difficult to read.

References Please see the order of the references, and how it should be presented, chronologically or alphabetically see several papers with same first author at different years. Is it authors surname and first name and later co-authors? please check.

Reference Din and Dai 1994 is not in the text.

Good job!

---

## Author Response (AR2)

Dear editor and two referees,

Thank you very much for your careful review and constructive suggestions regarding our manuscript "Early summer hydroclimatic signals were well captured by tree-ring earlywood width in the eastern Qinling Mountains, central China".

In the new revision, some tables and figures in the manuscript (Table 2 and 4, and Fig. 6–10) and supplementary (Table S1–S3, and Fig. S2–S4) have been modified due to the correction of running EPS and the subsequent change in tree-ring chronology length. However, these corrections did not affect our original conclusion.

Please see our point-by-point responses as below.

With best regards,

Yesi Zhao (on behalf of all co-authors)

**Response to the First Referee's Comments**

**1. Referee's comment:** I want to thank the editors the possibility to review again the article from Dr Yesi Zhao and colleagues and provide below comments and suggestions to improve the work presented. The actual version of the manuscript is much improved and therefore I can recommend it for publication after correcting some minor problems indicated below.

    **Author's Response:** We sincerely appreciate your comments on our revised manuscript. Please refer to the responses below. We hope you would be satisfied with our responses and revisions.

Just a few comments:

**2. Referee's comment:** I think that the signal obtained in EWW is rather alike to a signal of LWW. The measuring of EWW and LWW in Tsap involves some transformation of the data into individual time series. Can it be possible that the authors mixed up the conversion of the collected data?

    **Author's Response:** Thank you very much for this consideration. We rechecked the data and found sure that there was no mix of EWW and LWW.

    The similar signal shared by EWW and LWW may be due to their similar growth patterns. Here, we took the tree-ring width series detrended using negative exponential function together with linear regression with negative (or zero) slope (NELR) method as an example. As illustrated in Fig. 1, most of tree-ring samples showed significant correlations between their EWW and LWW. Therefore, the final EWW and LWW chronologies may contain similar signals even though EWW was more sensitive to early summer hydroclimate factors than LWW.

[Figure]

Figure 1. The Pearson's correlation coefficient between the EWW and LWW for each individual tree-ring sample (vertical axis) plotted against the corresponding degree of freedom (horizontal axis). The gray line denotes the correlation coefficients that are statistically significant at the 0.05 level based on two-tailed *t*-test. All EWW and LWW time series have been detrended using the NELR method.

**3. Referee's comment:** In addition, positive correlation with PDSI indicates that dry

conditions are favorable for tree growth. PDSI positive index indicates dry conditions while negative indicates wet conditions. I understand that the index uses both temperature and precipitation data and can be the signal of temperature that dominates the correlation with positive PDSI as it can be seen in figures 3 and 4. Either case, it is important the coverability between predictor and predictand, thus using tree-ring time series as predictor of PDSI, but if the signal is dominated by temperature rather than moisture, this can be the reason of the instability of the relationship of the time series indicated in the figure 8.

**Author's Response:** Thank you for these questions. In fact, the positive PDSI indicates a wet environment while the negative represents a dry condition (Palmer, 1965). Therefore, the positive correlation between PDSI (or scPDSI) and tree-ring width indicated that wet conditions favored tree growth.

Temperature might influence the instability of the relationship between scPDSI and EASMI showed in Fig. 8. However, comparison between scPDSI and temperature did not show enhanced correlation between scPDSI and temperature after the 1940s. Therefore, the cause for the weakened relationship is complicated and cannot be solely attributed to the impacts of temperature and precipitation. The specific cause is difficult to get because other factors such as available energy, humidity and wind speed can influence the scPDSI, too. We presented this discussion in the new revision, please refer to Line 27-32 Page 11 and Line 1-2 Page 12.

**4. Referee's comment:** Line 2 page 2, reconstructions cannot fall. Please reword.

**Author's Response:** We modified the "have fallen" into "have been conducted". Please refer to Line 2 Page 2.

**5. Referee's comment:** Line 7 page 2, I think in this case precipitation "are". Please check if I am correct.

**Author's Response:** According to the Collins English Dictionary (https://www.collinsdictionary.com/dictionary/english), the "precipitation" is an uncountable noun. The use of "is" can be found in Kidd and Levizzani (2001) as they expressed "……since precipitation is spatially and temporally highly variable ……".

**6. Referee's comment:** Line 11 page 2, please change unable for unsuitable.

**Author's Response:** Thank you for this suggestion. It has been modified. Please refer to Line 11 Page 2.

**7. Referee's comment:** Line 14 page 2, caused restrictions of water availability through the growing season

**Author's Response:** Thank you for this suggestion. It has been modified following your suggestion. Please refer to Line 14 Page 2.

**8. Referee's comment:** Line 18 page 2, are the abbreviations BYS and LCM needed? Since it is only one word for each name, I think is more reader friendly the original names, and using the locality names adds charm to the research presented.

**Author's Response:** Thank you for this suggestion. All abbreviations BYS and LCM in the manuscript have been modified into their full names.

**9. Referee's comment:** Line 20 page 2, please consider to mention SPEI index since it was added into the analysis and first mentioned in line 33.

**Author's Response:** Thank you for your suggestion. We have added the description about SPEI index in the introduction. Please refer to Line 26-29 Page 2.

**10. Referee's comment:** Line 22 page 2, please consider to give a very short background of the EASMI, what does it mean in terms of hydroclimate or temperature in the study area.

**Author's Response:** Thank you for your suggestion. We have added the contents about the background of EASMI and its impacts on the core region of EASM in the introduction. Besides, we removed the reasons that we selected this EASMI in the method section to avoid repetition. Please refer to Line 30 Page 2 to Line 3 Page 3.

**11. Referee's comment:** Materials and methods, Line 3 page 3. Please consider changing "study were provided by Shi et al. (2012)." By previously presented in Shi et al., 2012

**Author's Response:** Many thanks. We have modified this sentence following your suggestion. Please refer to Line 12 Page 3.

**12. Referee's comment:** Line 13 page 3: and did not complete around mid-Septermber. Please reword.

**Author's Response:** Thank you very much for pointing out this mistake. We have deleted the "r" from the mis-spelled "Septermber". Please refer to Line 23 Page 3.

**13. Referee's comment:** Line 13 page 3 "that the cambial cells of… started and ceased in… Please be so kind to indicate what started, cell division, cell growth, activity, etc.

**Author's Response:** Thank you for pointing out this unclear demonstration. It is the "activity" that cambial cell started. Please refer to Line 25 Page 3.

**14. Referee's comment:** Line 20 page 3, Due to the characteristics of the ring anatomy, Earlywood and latewood…

**Author's Response:** Many thanks. We have modified this sentence following your suggestion. Please refer to Line 29-30 Page 3.

**15. Referee's comment:** Line 10 page 4, please use "by means of" instead of "via" otherwise "through consecutive iterations"

**Author's Response:** Thank you for your suggestion. We used "by means of" instead of "via". Please refer to Line 20 Page 4.

**16. Referee's comment:** Line 12, see Osborn and colleagues, (1997).

**Author's Response:** We accept your suggestion. This error has been corrected in the

revised manuscript. Please refer to Line 22 Page 4.

**17. Referee's comment:** Line 13, EPS computed on multiple samples per tree will be inflated creating a wrong result (inflated value) please indicate if the index was calculated on single trees or multiple samples per tree.

    **Author's Response:** Thank you very much for the suggestion. In the original manuscript, we used the RCSsignalFree software to calculate the running Rbar and EPS based on the core numbers automatically. This did cause inflated EPS values due to the unequal number of cores per tree. In the new revision, we used the effective chronology signal (Rbar$_{eff}$) and the number of trees to calculate the running EPS with the function "rwi.stats.running" in R package "dplR" version 1.6.9 (Bunn et al., 2018). Please refer to Line 25-33 Page 4.

**18. Referee's comment:** Line 27 page 4. Please consider that the application of the principal component analysis is to reduce the amounts of variables of an analysis, rather than assess similarity between variables. Cluster analysis can be used to assess degrees of similarity between datasets.

    **Author's Response:** Thank you for your suggestion. In fact, we wanted to use the variance explained by the first eigenvector to estimate degree of similarity among detrended ring-width series just as used in Trouet et al. (2006). We reworded the "principal component" as "the variance explained by the first eigenvector", please refer to Line 6-7 Page 5.

**19. Referee's comment:** Line 2 page 5: SPEI and PDSI, please see the comment above on presenting both indices in the introduction.

    **Author's Response:** Thank you for your suggestion. Please refer to the response for **comment 9**.

**20. Referee's comment:** Line 5 page 5, please reword sentence "…and calculate the duration factors…" I do not understand what the authors mean.

    **Author's Response:** Thank you very much for your suggestion. We reworded the sentence as "Here we used the scPDSI instead of PDSI because it has solved the PDSI problems in spatial comparisons by calculating the duration factors (**weighting coefficients for the current moisture anomaly and the previous drought severity**) based on the characteristics of the climate at a given location (Wells et al., 2004)**"**. Please refer to Line 19 Page 5.

**21. Referee's comment:** Line 13 page 5, please be so kind to reword: three times the word "reconstruction" in one sentence makes it difficult to follow.

    **Author's Response:** Thank you very much for this suggestion. We changed the second "reconstruction" to "it", and the third to "time series". Please refer to Line 28 Page 5.

**22. Referee's comment:** Line 26 page 5, please remove "this" before EASMI and "is"

after it.

    **Author's Response:** Thank you for your suggestion. We moved the reason for choosing EASMI to the introduction, and this sentence was not presented in the new revision. Please refer to Line 30 Page 2 to Line 3 Page 3.

**23. Referee's comment:** Line 29 page 5. Please change "restored" for "adjusted"

    **Author's Response:** Many thanks. We have changed the word as your suggestion. Please refer to Line 4 Page 7.

**24. Referee's comment:** Line 10 page 7. Please change "manifest" for "assess"

    **Author's Response:** Thank you very much for this suggestion. The word has been modified. Please refer to Line 24 Page 7.

**25. Referee's comment:** Line 13 page 7, please provide proper citation and names for software Matlab RM and Surfer 10.

    **Author's Response:** Thank you for your suggestion. We have added the version of Matlab, and cited the corresponding publisher for the two software. Please refer to Line 27-28 Page 7.

**26. Referee's comment:** Line 24 page 7. The positive correlation with PDSI indicates dry conditions as favorable for tree growth. Do I get that right? This means that conditions in the study area are rather too wet, and a moderate drought will be good the tree growth. Is this reasonable for an area of thin soils and well drained?

    **Author's Response:** Please refer to the response for **comment 3**.

**27. Referee's comment:** Line 15 page 8, it says "from September of two years earlier to May of last May". Please reword.

    **Author's Response:** Thank you for pointing out this mistake. We have corrected the "May of last May" into "May of the previous year". Please refer to Line 30-31 Page 8.

**28. Referee's comment:** Line 5-6 page 9, please reword "Increased water deficiency due to the rising temperature and inadequate rainfall in the 5 early growing season could induce soil water deficiency thus suppressing cell expansion and cell growth in the cambium"

    **Author's Response:** As in the response for **comment 3**, the positive relationship between EWW and scPDSI did indicate a water restriction on tree growth. Therefore, we think the reason we presented here is reasonable. To be consistent with the demonstration in Fritts (1976), we reword the "soil water deficiency" as "water stress", and the "cell expansion and cell growth in the cambium" as "cell division and expansion". Please refer to Line 21-22, Page 9.

**29. Referee's comment:** Line 22, page 9. Restored or adjusted?

    **Author's Response:** Thank you for this suggestion. We used "adjusted" instead of "restored". Please refer to Line 4 Page 10.

**30. Referee's comment:** Line 23 page 9, please add "the" before reconstruction.

   **Author's Response:** Thank you for this suggestion. We have added "the" before "reconstruction". Please refer to Line 5 Page 10.

**31. Referee's comment:** Line 6 page 10. Please change "The possible reasons might be that (1) the seasons that the three reconstructions aimed at are different from ours…" for "The possible reasons might be that (1) the reconstructions aim to different target season"

   **Author's Response:** Thank you for this suggestion. We have modified this sentence following your suggestion. Please refer to Line 20 Page 10.

**32. Referee's comment:** Line 33 page 10: How stable in time is the spatial pattern of the EASM?

   **Author's Response:** Thank you for raising this question. Just as demonstrated in Wang et al., (2008), the EASM is a non-stationary system and could change on interdecadal timescale and may even much longer timescales. Please refer to the revision in Line 13-15 Page 11.

**33. Referee's comment:** Line 25 page 11, please reword and eliminate "etc,"

   **Author's Response:** Thank you for this suggestion. It was done. Please refer to Line 14 Page 12

**Figures:**
**34. Referee's comment:** Figure 1, Southern Asia summer monsoon is not mentioned in the text

   **Author's Response:** Thank you for pointing out this issue. Since the climate of the study area is governed by the East Asian summer monsoon (EASM) as indicated by the boundary of EASM (Please refer to Fig. 1), we focused on the relationship of our reconstruction with EASM in this study. Of course, just as your valuable consideration, the Southern Asian summer monsoon (SASM) may impact our study area (Zhang, 2001). We would analysis the relationship of our reconstruction with SASM, and other large-scale circulation patterns including PDO, AMO, etc in the future.

**35. Referee's comment:** Figure 2 figure text: please add "s" to denote.

   **Author's Response:** Thank you very much for pointing out this mistake. It has been modified.

**36. Referee's comment:** Figure 6 add "d" to shade. Areas

   **Author's Response:** Thank you very much for pointing out this mistake. It has been modified.

**37. Referee's comment:** Figure 9, figure text, citation Macias-Faura et al.",",2012) please check for the right format.

**Author's Response:** Thank you for reminding. The comma is needed as required by the Journal.

**38. Referee's comment:** Figure 10, text in the figure: Huaihe River and Yangtze River in the lower left plate is difficult to read.
   **Author's Response:** Thank you very much for this suggestion. We have added a white border for each character to make them more recognizable.

**39. Referee's comment:** References Please see the order of the references, and how it should be presented, chronologically or alphabetically see several papers with same first author at different years. Is it authors surname and first name and later co-authors? please check.
   **Author's Response:** Many thanks. In this revision, we have checked the order of reference carefully. The found mistakes have been corrected.

**40. Referee's comment:** Reference Din and Dai 1994 is not in the text.
   **Author's Response:** There is no "Din and Dai 1994" in the reference list.

**41. Referee's comment:** Good job!
   **Author's Response:** Thanks again for your positive comments on our work.

We thoroughly checked the time intervals again in the revision. Thanks again for your consideration.

[revised manuscript text omitted]

**Supplement**

**Table S1.** Correlation coefficients between the tree–ring width chronologies from the two study sites, Baiyunshan and Longchiman, over their common period 1850–2005.

| Chronologies | Detrending method | | |
|---|---|---|---|
| | NELR | SP67 | SPA50 |
| EWW STD | 0.63 / 0.59 | 0.64 / 0.60 | 0.62 / 0.57 |
| LWW STD | 0.40 / 0.51 | 0.46 / 0.49 | 0.46 / 0.47 |
| TRW STD | 0.58 / 0.58 | 0.60 / 0.62 | 0.61 / 0.63 |
| EWW SSF | 0.64 / 0.60 | 0.57 / 0.53 | 0.55 / 0.49 |
| LWW SSF | 0.43 / 0.54 | 0.37 / 0.51 | 0.35 / 0.47 |
| TRW SSF | 0.57 / 0.57 | 0.50 / 0.54 | 0.50 / 0.54 |

Note: The correlation coefficients before (after) the slashes are for the original (prewhitened and linearly detrended) STD and SSF chronologies. All correlation coefficients are statistically significant at the 0.001 level based on Monte Carlo test (Efron and Tibshirani, 1986; Macias-Fauria et al., 2012).

删除了: BYS…aiyunshan and LongchimanLCM… over their common period …

删除了: when EPS larger than 0.85.

删除了: 8 (1888)…/ 0.5978 (1889)

删除了: 82 (1888)…/ 0.60.8 (1889)

删除了: 82 (1888)…/ 0.57.8 (1889)

删除了: 65 (1892)…/ 0.5167…(1897)

删除了: 73 (1907)…/ 0.4969 (1912)

删除了: 67 (1892)…/ 0.4763 (1897)

删除了: 76 (1884)…/ 0.5872 (1888)

删除了: 8 (1887)…/ 0.6276 (1892)

删除了: 80 (1887)…/ 0.6376 (1892)

删除了: 79 (1888)…/ 0.6078 (1889)

删除了: 76 (1888) … 0.5377 (1889)

删除了: 81 (1888)…/ 0.4978 (1889)

删除了: 67 (1892)…/ 0.5468 (1897)

删除了: 58 (1892)…/ 0.5162 (1897)

删除了: 63 (1896)…/ 0.4764…(1901)

删除了: 74 (1884)…/ 0.5772 (1888)

删除了: 74 (1887)…/ 0.5473 (1892)

删除了: 76 (1884)…/ 0.5474 (1889)

删除了: . The beginning year of the common period is shown in the bracket. The ending year for all chronologies is 2005.

**Table S2.** Descriptive statitics of the composite STD and SSF tree–ring width chronologies when EPS ≧0.85.

| Detrending method | Chronology | Starting year when EPS ≧ 0.85 | Standard deviation | Mean sensitivity | First-order autocorrelation |
|---|---|---|---|---|---|
| NELR | EWW STD | 1868 | 0.238 | 0.225 | 0.348 |
| | LWW STD | 1877 | 0.25 | 0.221 | 0.421 |
| | TRW STD | 1871 | 0.221 | 0.200 | 0.458 |
| SP67 | EWW STD | 1871 | 0.235 | 0.222 | 0.354 |
| | LWW STD | 1877 | 0.247 | 0.227 | 0.38 |
| | TRW STD | 1871 | 0.218 | 0.200 | 0.439 |
| SPA50 | EWW STD | 1867 | 0.234 | 0.223 | 0.328 |
| | LWW STD | 1875 | 0.245 | 0.225 | 0.384 |
| | TRW STD | 1866 | 0.215 | 0.200 | 0.42 |
| NELR | EWW SSF | 1868 | 0.245 | 0.226 | 0.382 |
| | LWW SSF | 1875 | 0.263 | 0.221 | 0.451 |
| | TRW SSF | 1868 | 0.227 | 0.200 | 0.479 |
| SP67 | EWW SSF | 1868 | 0.245 | 0.225 | 0.397 |
| | LWW SSF | 1875 | 0.255 | 0.226 | 0.404 |
| | TRW SSF | 1871 | 0.227 | 0.201 | 0.48 |
| SPA50 | EWW SSF | 1867 | 0.234 | 0.207 | 0.418 |
| | LWW SSF | 1875 | 0.246 | 0.214 | 0.426 |
| | TRW SSF | 1866 | 0.214 | 0.191 | 0.459 |

Note: All statistics were calculated using the R package "dplR" version 1.6.9 (Bunn et al., 2018).

删除了: ⋯ >
删除了: 0.232
删除了: 0.306
设置了格式
设置了格式
删除了: 4
删除了: 17
设置了格式
删除了: 0.25
设置了格式
删除了: 0.358
删除了: 1
删除了: 19
删除了: 0.216
设置了格式
设置了格式
删除了: 0.389
删除了: 64
删除了: 194
删除了: 0.232
设置了格式
设置了格式
删除了: 0.29
删除了: 64
删除了: 18
设置了格式
删除了: 0.249
删除了: 0.313
设置了格式
删除了: 1
删除了: 9
设置了格式
删除了: 0.215
设置了格式
删除了: 0.363
删除了: 64
删除了: 196
设置了格式
删除了: 0.232
设置了格式
删除了: 0.281
删除了: 4
删除了: 218
设置了格式
设置了格式

**Table S3.** Statitics of the detrended ring-width series over their common period 1915–2005.

| Detrending method | Chronology | $Var_{pc1}$ | $Rbar_{eff}$ | SNR | EPS |
|---|---|---|---|---|---|
| NELR | EWW STD | 0.386 | 0.443 | 25.878 | 0.963 |
| | LWW STD | 0.328 | 0.358 | 18.175 | 0.948 |
| | TRW STD | 0.384 | 0.433 | 24.806 | 0.961 |
| SP67 | EWW STD | 0.427 | 0.492 | 31.452 | 0.969 |
| | LWW STD | 0.353 | 0.400 | 21.702 | 0.956 |
| | TRW STD | 0.422 | 0.481 | 30.104 | 0.96 |
| SPA50 | EWW STD | 0.411 | 0.471 | 29.007 | 0.967 |
| | LWW STD | 0.341 | 0.384 | 20.315 | 0.953 |
| | TRW STD | 0.404 | 0.459 | 27.607 | 0.965 |
| NELR | EWW SSF | 0.399 | 0.456 | 27.291 | 0.965 |
| | LWW SSF | 0.317 | 0.346 | 17.252 | 0.945 |
| | TRW SSF | 0.393 | 0.439 | 25.492 | 0.962 |
| SP67 | EWW SSF | 0.453 | 0.520 | 35.298 | 0.972 |
| | LWW SSF | 0.366 | 0.417 | 23.308 | 0.959 |
| | TRW SSF | 0.441 | 0.504 | 33.033 | 0.971 |
| SPA50 | EWW SSF | 0.474 | 0.539 | 37.994 | 0.974 |
| | LWW SSF | 0.352 | 0.399 | 21.627 | 0.956 |
| | TRW SSF | 0.431 | 0.491 | 31.354 | 0.969 |

Note: The common period for the tree–ring width dataset was calculated with the "common.interval" function in R package "dplR" version 1.6.9 (Bunn et al., 2018) based on a trade–off between the maximum number of series and years. The statistics, $Var_{pc1}$, $Rbar_{eff}$, SNR, and EPS represent the variance explained by the first eigenvector, effective chronology signal, signal–to–noise ratio, and expressed population signal, respectively. The $Var_{pc1}$ was calculated using the Program ARSTAN40c (Cook and Krusic, 2006), and the other statistics were calculated using the R package "dplR" version 1.6.9 (Bunn et al., 2018).

**Table S4.** The meterological stations utilized by CRU dataset (http://www.cru.uea.ac.uk/data) located

in the area between latitudes 32° N and 34.5° N, and longitudes 111° E and 112° E.

| Meteorological station | Longitude (°E) | Latitude (°N) | Climatic factor | Temporal cover |
|---|---|---|---|---|
| Lushi | 111.03 | 34.05 | Precipitation | 1952.07–2013.12 |
| | | | Temperature | 1952.07–2016.12 |
| Laohekou | 111.73 | 32.43 | Precipitation | 1933.02–1933.11; 1934.01–1935.06; 1935.08–1935.12; 1936.08–1938.07; 1950.06–2005.06; 2005.08–2013.12 |
| | | | Temperature | 1951.01–2016.12 |
| Yunxian | 111.8 | 32.9 | Precipitation | 1933.03–1938.05; 1938.07–1947.10; 1950.03–1990.12; 1991.05; 1991.07–1991.08 |

删除了: e

[Figure]

**Figure S1.** Scanned photograph of a section of *P. tabulaeformis* tree–ring sample (LCM0118A). The distinct earlywood (EW) and latewood (LW) segments can be identified by inspection under a microscope.

[Figure]

[Figure]

删除了：

**Figure S2.** Standard tree-ring width chronologies (solid curve) generatued using three kinds of detrending methods for **(a–c)** earlywood width (EWW), **(d–f)** latewood width (LWW), and **(g–i)** total tree–ring width (TRW) at the two study sites, Baiyunshan (black) and Longchiman (red). The three kinds of detrending methods are: (1) negative exponential function together with linear regession with negative (or zero) slope (NELR), (2) cubic smoothed splines with a 50 % frequency cutoff of 67 % of the series length (SP67), and (3) age-dependent splines with an initial stiffness of 50 years (SPA50). The dashed and dotted curves denote the running expressed population signal (EPS) and effective chronology signal (Rbar_eff), respectively. The horizontal line indicates the threshold EPS value of 0.85. The running EPS and Rbar_eff values were calculated over a 51–year window.

删除了：YS

删除了：LCM

设置了格式：下标

删除了：values

删除了：Rbar

[Figure]

[Figure]

删除了：

**Figure S3.** The same as Figure S2, but for the signal-free (SSF) chronologies.

[Figure]

[Figure]

删除了：

**Figure S4.** Composite **(a–c)** STD and **(d–f)** SSF tree-ring width chronologies for EWW (red), LWW (black), and TRW (blue) generated using the merged tree–ring samples from the two study sites, Baiyunshan and Longchiman, based on three kinds of detrending methods. The detrending methods are: (1) negative exponential function together with linear regression with negative (or zero) slope (NELR), (2) cubic smoothed splines with a 50 % frequency cutoff of 67 % of the series length (SP67), and (3) age-dependent splines with an initial stiffness of 50 years (SPA50). The dashed and dotted curves denote the running expressed population signal (EPS) and effective chronology signal ($Rbar_{eff}$), respectively. The horizontal line indicates the threshold EPS value of 0.85. The running EPS and $Rbar_{eff}$ values were calculated over a 51–year window. The segement plot indicated the sample depth (core).

删除了：BYS

设置了格式：字体：(默认) Times New Roman, (中文) Times New Roman

设置了格式：字体：(默认) Times New Roman, (中文) Times New Roman

设置了格式：字体：(默认) Times New Roman, (中文) Times New Roman

删除了：CM

删除了：Rbar values,

删除了：Rbar

[Figure]

**Figure S5.** Spatial distribution of the meteorological stations included in the CRU dataset (http://www.cru.uea.ac.uk/data) around the tree-ring sampling sites, Baiyunshan and Longchiman. Black cycles represent the meteorological stations that provide both precipitation and temperature data. While, blue (yellow) cycles represent those that only provide precipitation (temperature) data. The dashed rectangle indicates the range of scPDSI grid points used for calibration in this study. The color patches denote the spatial correlation coefficients between May-July scPDSI and NELR based EWW STD chronology.

删除了: BYS

删除了: CM

[Figure]

**Figure S6.** Correlation coefficients between the tree-ring width chronologies and multi-month averaged

scPDSI (April to September of the current year). The up triangle, down triangle, and circle indicate the

chronologies of EWW, LWW, and TRW, respectively. The color black, red and blue indicate the STD

chronologies generated using the detrending methods NELR, SP67, and SPA50, respectively. The color

magenta, cyan, and orange indicate the SSF chronologies generated using the detrending methods NELR,

SP67, and SPA50, respectively. The dashed symbol indicates that the correlation does not reach the 0.05

significance level, which were tested using the Monte Carlo method (Efron and Tibshirani, 1986; Macias-

Fauria et al., 2012).

[Figure]

**Figure S7.** Comparisons between the MJJ scPDSI (gray) and EWW STD chronologies generated using the detrending methods NELR (black), SP67 (red), and SPA50 (blue) during the period 1953–2005. The corresponding linear trends are indicated by the dashed lines, and the slope statistics are labelled in the bottom left corner of the figure.

删除了: were

删除了: were

---

## Author Response (AR3)

**Response to the Editor's Comments**

**1. Editor's comment:** Motivate (briefly) why EW is suitable for your reconstruction, e.g. by informing when the EW is formed..

   **Author's Response:** Thank you very much for the suggestion. We added the formation period of EWW in the *Introduction* section. Please refer to Line10 on Page 3.

**2. Editor's comment:** The reconstruction is "biased" towards wet values, especially in its early part (see e.g. table 4). This is not well reflected in the other hydroclimate reconstructions you compare with, and the EWW data does not seem to fully capture the observed dry years. Is this real or due to the data? It could be good to briefly discuss this point.

   **Author's Response:** Thank you for the question. One possible reason is that the reconstruction has underestimated the drought severity in its early part. This may be due to the low-frequency hydroclimatic signals being removed during the detrending process when using simple curve fitting methods (Briffa et al., 1996). The Regional Curve Standardization (RCS) method might preserve more low-frequency signals (Briffa et al., 1992), but it requires more filed work. We added the relevant discussion in Line 7-13 on Page 11.

**3. Editor's comment:** To make the paper more accessible, you should consider letting a native speaker go through the text. It is generally very well written, but still there is room for improvement.

   **Author's Response:** Thank you for suggestion. We asked a professional English editing company to improve the English writing completely. The *editorial certificate* is attached as follows.

[Figure]

**4. Editor's comment:** Note that the figures should be able to stand on their own, so all necessary information (including relevant references) need to be in the figure captions.

    **Author's Response:** Thank you for the suggestion. We rechecked all the figure captions again. A complete caption description was added for Fig.4.

**5. Editor's comment:** In the first section you refer to Liu et al. 2018b before 2018a, please change and have a general look at the references throughout.

    **Author's Response:** Thank you for the suggestion. We sorted the reference using the LaTex package provided by the journal CP. "*Liu, X et al. 2018* (*Liu et al., 2018a*)" was automatically put before "*Liu, Y et al., 2018* (*Liu et al., 2018b*)" in the reference list. Since we cited "*Liu, Y et al., 2018* (*Liu et al., 2018b*)" firstly and "*Liu, X et al. 2018* (*Liu et al., 2018a*)" secondly in the text, "*Liu et al., 2018b*" naturally came in front of "*Liu et al., 2018a*" .

**6. Editor's comment:** I strongly recommend that you make the data presented in this paper available through an open repository rather than "on request", at least the data plotted in Fig. 6c.

    **Author's Response:** Thank you for the suggestion. We added the MJJ scPDSI reconstruction in the supplementary materials.

**Note:** The regional annual mean precipitation (Line 20, Page 3) and the Figs. 2, 3 and 4 were updated because we corrected the average of precipitation data. According to Jones and Hulme (1996), regional precipitation series can be calculated by firstly deriving the regional averages in terms of percentages, then multiplying the regional mean to transform the resulting series back to millimeter units. In the previous versions, however, we misused "standard deviation" rather than "mean" as the denominator to calculate percentages. The correction has an influence on the regional annual and monthly mean precipitation amount, but a very slight impact on precipitation variation and its correlation with tree-ring width series. The correction does not change the results and conclusions.

    Thank you again for your careful review and valuable suggestions for the manuscript.

Reference:

Briffa, K. R., Jones, P. D., Bartholin, T. S., Eckstein, D., Schweingruber, F. H., Karlén, W., Zetterberg, P., and Eronen, M.: Fennoscandian summers from AD 500: temperature changes on short and long timescales, Clim. Dynam., 7, 111–119, https://doi.org/10.1007/BF00211153, 1992.

Briffa, K. R., Jones, P. D., Schweingruber, F. H., Karlén, W., and Shiyatov, S. G.: Tree-ring variables as proxy-climate indicators: problems with low-frequency signals, in: Climatic variations and forcing mechanisms of the last 2000 years, edited by: Jones, P. D., Bradley, R. S., and Jouzel, J., Springer, Berlin, Heidelberg, 9–41, http:/doi.org/10.1007/978-3-642-61113-1_2, 1996.

Jones, P. D., and Hulme, M.: Calculating regional climatic time series for temperature and precipitation: methods and illustrations, Int. J. Climatol., 16, 361–377, https://doi.org/10.1002/(SICI)1097-0088(199604)16:4<361::AID-JOC53>3.0.CO;2-F, 1996.

**Early summer hydroclimatic signals are well captured by tree-ring earlywood width in the eastern Qinling Mountains, central China**

Yesi Zhao[1,2], Jiangfeng Shi[1,3], Shiyuan Shi[1], Xiaoqi Ma[1], Weijie Zhang[1], Bowen Wang[1], Xuguang Sun[4], Huayu Lu[1], Achim Bräuning[2]

[1] School of Geography and Ocean Science, Nanjing University, Nanjing 210023, China
[2] Institute of Geography, Friedrich-Alexander-University Erlangen-Nürnberg, Erlangen 91058, Germany
[3] Laboratory of Tree-Ring Research, University of Arizona, Tucson 85721, USA
[4] School of Atmospheric Sciences, Nanjing University, Nanjing 210023, China

*Correspondence to*: Jiangfeng Shi (shijf@nju.edu.cn)

**Abstract.** In the humid and semi-humid regions of China, tree-ring width (TRW) chronologies offer limited moisture-related climatic information. To gather additional climatic information, it would be interesting to explore the potentials of the intra-annul tree-ring width indices (i.e., the earlywood width (EWW) and latewood width (LWW)). To achieve this purpose, TRW, EWW and LWW were measured from the tree-ring samples of *Pinus tabulaeformis* originating from the semi-humid eastern Qinling Mountains, central China. Standard (STD) and signal-free (SSF) chronologies of all parameters were created using these detrending methods including (1) negative exponential functions combined with linear regression with negative (or zero) slope (NELR), (2) cubic smoothing splines with a 50 % frequency cutoff at 67 % of the series length (SP67), and (3) age-dependent splines with an initial stiffness of 50 years (SPA50). The results showed that EWW chronologies were significantly negatively correlated with temperature, but positively correlated with precipitation and soil moisture conditions during the current early growing season. By contrast, LWW and TRW chronologies had weaker relationships with these climatic factors. The strongest climatic signal was detected for the EWW STD chronology detrended with the NELR method, explaining 50 % of the variance of the May–July self-calibrated Palmer Drought Severity Index (MJJ scPDSI) during the instrumental period 1953–2005. Based on this relationship, the MJJ scPDSI was reconstructed back to 1868 using a linear regression function. The reconstruction was validated by comparison with other hydroclimatic reconstructions and historical document records from adjacent regions. Our results highlighted the potentials of intra-annual tree-ring indices for reconstructing seasonal hydroclimatic variations in humid and semi-humid regions of China. Furthermore, our reconstruction exhibited a strong in-phase relationship with a newly proposed East Asian summer monsoon index (EASMI) before the 1940s on the decadal and longer timescales, which may be due to the positive response of the local precipitation to EASMI. Nonetheless, the cause for the weakened relationship after the 1940s is complex, and cannot be solely attributed to the changing impacts of precipitation and temperature.

删除了: were
删除了: T
删除了: could only provide
删除了: amount of
删除了: in the humid and semi-humid regions of China;
删除了: thus,
删除了: is
删除了: worth
删除了: to
删除了: provide some additional climatic information. To fulfil this task,
删除了: in a semi-humid region
删除了: , that is, the
删除了: Their s
删除了: created
删除了: different
删除了: together
删除了: smoothed
删除了: of
删除了: Comparatively
删除了: with
删除了: detrending method of
删除了: contained the strongest climatic signal
删除了: t
删除了: further
删除了: ng
删除了: in
删除了: This
删除了: reconstruction
删除了: the
删除了: However
删除了: complicated

[revised manuscript text omitted]
}[\text{u}(2.5°–10°\,\text{N}, 105°–140°\,\text{E}) − \text{u}(17.5°–22.5°\,\text{N}, 105°–140°\,\text{E}) + \text{u}(30°–37.5°\,\text{N}, 105°–140°\,\text{E})] \quad (1)$$

where Nor and u are standardization and mean 200 hPa zonal wind, respectively. To understand the possible impacts of local precipitation and temperature (32° –34.5° N and 111° –112° E) on the relationship between the scPDSI and EASMI, the precipitation and temperature data were extracted from the gridded precipitation dataset Global Precipitation Climatology Centre Version 7 (GPCC v7; Schneider et al., 2015), and gridded temperature dataset Climatic Research Unit Time-Series Version 4.01 (CRU TS 4.01; Harris et al., 2014), respectively. The gridded dataset can represent the variations of precipitation and temperature over East China in the 20$^{th}$ century (Wang and Wang, 2017; Wen et al., 2006).

**2.5 Statistical methods**

To investigate the climate response of different tree-ring parameters (EWW, LWW, and TRW), we first calculated the Pearson correlation coefficients of the STD and SSF tree-ring width chronologies using monthly climate time series. The time window for the correlation analysis spanned from January of two years before tree-ring formation to October of the current growth year. Next, correlations were calculated between the prewhitened and linearly detrended chronologies and climate time series in order to evaluate the possible effects of autocorrelations and secular trends. The prewhitening procedure was performed using the "ar" function in R package "stats" version 3.5.1 (R Core Team, 2018). The appropriate autoregressive order was automatically determined by the Akaike Information Criterion (Akaike, 1974). The linear detrending procedure was performed based on the "detrend" function in Matlab R2016a (The MathWorks, Inc., 2016). To find the strongest climate-growth relationship, we analyzed the response of different tree-ring parameters to multi-month averaged scPDSI (which had the stronger impacts on tree-growth than other climatic factors; see the results for details). Finally, we adopted the wavelet coherence method (Grinsted et al., 2004) to test the temporal stability and possible lags of the climate-growth relationship on different frequency domains.

[revised manuscript text omitted]

**Supplementary material**

**Table S1.** Correlation coefficients between the tree–ring width chronologies from the two study sites, Baiyunshan and Longchiman, over their common period 1850–2005.

| Chronologies | Detrending method | | |
|---|---|---|---|
| | NELR | SP67 | SPA50 |
| EWW STD | 0.63 / 0.59 | 0.64 / 0.60 | 0.62 / 0.57 |
| LWW STD | 0.40 / 0.51 | 0.46 / 0.49 | 0.46 / 0.47 |
| TRW STD | 0.58 / 0.58 | 0.60 / 0.62 | 0.61 / 0.63 |
| EWW SSF | 0.64 / 0.60 | 0.57 / 0.53 | 0.55 / 0.49 |
| LWW SSF | 0.43 / 0.54 | 0.37 / 0.51 | 0.35 / 0.47 |
| TRW SSF | 0.57 / 0.57 | 0.50 / 0.54 | 0.50 / 0.54 |

Note: The correlation coefficients before (after) the slashes are for the original (prewhitened and linearly detrended) STD and SSF chronologies. All correlation coefficients are statistically significant at the 0.001 level based on Monte Carlo test (Efron and Tibshirani, 1986; Macias-Fauria et al., 2012).

**Table S2.** Descriptive statitics of the composite STD and SSF tree–ring width chronologies when EPS

≥0.85.

| Detrending method | Chronology | Starting year when EPS ≥ 0.85 | Standard deviation | Mean sensitivity | First-order autocorrelation |
|---|---|---|---|---|---|
| NELR | EWW STD | 1868 | 0.238 | 0.225 | 0.348 |
| | LWW STD | 1877 | 0.25 | 0.221 | 0.421 |
| | TRW STD | 1871 | 0.221 | 0.200 | 0.458 |
| SP67 | EWW STD | 1871 | 0.235 | 0.222 | 0.354 |
| | LWW STD | 1877 | 0.247 | 0.227 | 0.38 |
| | TRW STD | 1871 | 0.218 | 0.200 | 0.439 |
| SPA50 | EWW STD | 1867 | 0.234 | 0.223 | 0.328 |
| | LWW STD | 1875 | 0.245 | 0.225 | 0.384 |
| | TRW STD | 1866 | 0.215 | 0.200 | 0.42 |
| NELR | EWW SSF | 1868 | 0.245 | 0.226 | 0.382 |
| | LWW SSF | 1875 | 0.263 | 0.221 | 0.451 |
| | TRW SSF | 1868 | 0.227 | 0.200 | 0.479 |
| SP67 | EWW SSF | 1868 | 0.245 | 0.225 | 0.397 |
| | LWW SSF | 1875 | 0.255 | 0.226 | 0.404 |
| | TRW SSF | 1871 | 0.227 | 0.201 | 0.48 |
| SPA50 | EWW SSF | 1867 | 0.234 | 0.207 | 0.418 |
| | LWW SSF | 1875 | 0.246 | 0.214 | 0.426 |
| | TRW SSF | 1866 | 0.214 | 0.191 | 0.459 |

Note: All statistics were calculated using the R package "dplR" version 1.6.9 (Bunn et al., 2018).

**Table S3.** Statitics of the detrended ring-width series over their common period 1915–2005.

| Detrending method | Chronology | $Var_{pc1}$ | $Rbar_{eff}$ | SNR | EPS |
|---|---|---|---|---|---|
| NELR | EWW STD | 0.386 | 0.443 | 25.878 | 0.963 |
| | LWW STD | 0.328 | 0.358 | 18.175 | 0.948 |
| | TRW STD | 0.384 | 0.433 | 24.806 | 0.961 |
| SP67 | EWW STD | 0.427 | 0.492 | 31.452 | 0.969 |
| | LWW STD | 0.353 | 0.400 | 21.702 | 0.956 |
| | TRW STD | 0.422 | 0.481 | 30.104 | 0.96 |
| SPA50 | EWW STD | 0.411 | 0.471 | 29.007 | 0.967 |
| | LWW STD | 0.341 | 0.384 | 20.315 | 0.953 |
| | TRW STD | 0.404 | 0.459 | 27.607 | 0.965 |
| NELR | EWW SSF | 0.399 | 0.456 | 27.291 | 0.965 |
| | LWW SSF | 0.317 | 0.346 | 17.252 | 0.945 |
| | TRW SSF | 0.393 | 0.439 | 25.492 | 0.962 |
| SP67 | EWW SSF | 0.453 | 0.520 | 35.298 | 0.972 |
| | LWW SSF | 0.366 | 0.417 | 23.308 | 0.959 |
| | TRW SSF | 0.441 | 0.504 | 33.033 | 0.971 |
| SPA50 | EWW SSF | 0.474 | 0.539 | 37.994 | 0.974 |
| | LWW SSF | 0.352 | 0.399 | 21.627 | 0.956 |
| | TRW SSF | 0.431 | 0.491 | 31.354 | 0.969 |

Note: The common period for the tree–ring width dataset was calculated with the "common.interval" function in R package "dplR" version 1.6.9 (Bunn et al., 2018) based on a trade–off between the maximum number of series and years. The statistics, $Var_{pc1}$, $Rbar_{eff}$, SNR, and EPS represent the variance explained by the first eigenvector, effective chronology signal, signal–to–noise ratio, and expressed population signal, respectively. The $Var_{pc1}$ was calculated using the Program ARSTAN40c (Cook and Krusic, 2006), and the other statistics were calculated using the R package "dplR" version 1.6.9 (Bunn et al., 2018).

**Table S4.** The meterological stations utilized by CRU dataset (http://www.cru.uea.ac.uk/data) located

in the area between latitudes 32° N and 34.5° N, and longitudes 111° E and 112° E.

| Meteorological station | Longitude (°E) | Latitude (°N) | Climatic factor | Temporal cover |
|---|---|---|---|---|
| Lushi | 111.03 | 34.05 | Precipitation | 1952.07–2013.12 |
| | | | Temperature | 1952.07–2016.12 |
| Laohekou | 111.73 | 32.43 | Precipitation | 1933.02–1933.11; 1934.01–1935.06; 1935.08–1935.12; 1936.08–1938.07; 1950.06–2005.06; 2005.08–2013.12 |
| | | | Temperature | 1951.01–2016.12 |
| Yunxian | 111.8 | 32.9 | Precipitation | 1933.03–1938.05; 1938.07–1947.10; 1950.03–1990.12; 1991.05; 1991.07–1991.08 |

**Table S5.** The reconstructed May–July (MJJ) scPDSI during the period 1868–2005.

| Year | scPDSI | Year | scPDSI | Year | scPDSI | Year | scPDSI |
|---|---|---|---|---|---|---|---|
| 1868 | -1.427 | 1904 | 1.167 | 1940 | -1.139 | 1976 | -1.394 |
| 1869 | 2.307 | 1905 | 2.260 | 1941 | -1.300 | 1977 | -1.474 |
| 1870 | -0.388 | 1906 | 3.654 | 1942 | 0.859 | 1978 | -1.829 |
| 1871 | -0.321 | 1907 | -0.630 | 1943 | 1.046 | 1979 | -0.046 |
| 1872 | -0.227 | 1908 | -0.080 | 1944 | 2.380 | 1980 | 2.065 |
| 1873 | -0.918 | 1909 | -0.777 | 1945 | -0.656 | 1981 | 0.088 |
| 1874 | -1.682 | 1910 | 2.045 | 1946 | 1.589 | 1982 | 0.859 |
| 1875 | -1.374 | 1911 | 3.842 | 1947 | 0.986 | 1983 | 4.150 |
| 1876 | 0.564 | 1912 | 2.012 | 1948 | 2.809 | 1984 | 2.327 |
| 1877 | -1.568 | 1913 | -0.495 | 1949 | 2.950 | 1985 | 1.891 |
| 1878 | 0.570 | 1914 | -0.006 | 1950 | 1.998 | 1986 | -0.770 |
| 1879 | -3.612 | 1915 | 0.939 | 1951 | 1.428 | 1987 | 1.851 |
| 1880 | -1.984 | 1916 | -0.636 | 1952 | 1.549 | 1988 | -0.127 |
| 1881 | -0.991 | 1917 | -1.179 | 1953 | 1.160 | 1989 | 0.269 |
| 1882 | 1.958 | 1918 | -0.710 | 1954 | 1.355 | 1990 | 2.320 |
| 1883 | 2.622 | 1919 | -0.167 | 1955 | -1.682 | 1991 | 1.194 |
| 1884 | 1.254 | 1920 | -1.749 | 1956 | 1.415 | 1992 | -0.991 |
| 1885 | 3.071 | 1921 | 0.497 | 1957 | 0.356 | 1993 | 1.288 |
| 1886 | 1.522 | 1922 | 0.758 | 1958 | 0.550 | 1994 | -2.118 |
| 1887 | 1.321 | 1923 | -2.279 | 1959 | 0.523 | 1995 | -2.104 |
| 1888 | 0.624 | 1924 | -1.072 | 1960 | -0.107 | 1996 | 0.041 |
| 1889 | 1.254 | 1925 | 0.255 | 1961 | -0.247 | 1997 | -0.556 |
| 1890 | 0.624 | 1926 | -2.325 | 1962 | 0.517 | 1998 | 1.341 |
| 1891 | -1.441 | 1927 | -0.341 | 1963 | 0.336 | 1999 | 0.115 |
| 1892 | -0.743 | 1928 | -1.782 | 1964 | 1.616 | 2000 | -2.942 |
| 1893 | 1.462 | 1929 | -2.527 | 1965 | -0.241 | 2001 | -1.963 |
| 1894 | 3.057 | 1930 | 1.931 | 1966 | 0.229 | 2002 | 0.289 |
| 1895 | 2.112 | 1931 | 1.495 | 1967 | 0.262 | 2003 | 0.624 |
| 1896 | 1.549 | 1932 | -0.569 | 1968 | -1.568 | 2004 | 0.517 |
| 1897 | 1.120 | 1933 | 2.514 | 1969 | -0.878 | 2005 | 1.777 |
| 1898 | 3.768 | 1934 | 3.111 | 1970 | 1.355 | | |
| 1899 | -1.025 | 1935 | 0.523 | 1971 | 1.777 | | |
| 1900 | -2.238 | 1936 | 3.721 | 1972 | 0.839 | | |
| 1901 | -0.489 | 1937 | 0.423 | 1973 | 2.970 | | |
| 1902 | -1.923 | 1938 | 0.249 | 1974 | 1.428 | | |
| 1903 | 0.544 | 1939 | -1.018 | 1975 | 1.020 | | |

设置了格式: 英语(英国)

[Figure]

**Figure S1.** Photograph of a section of a *P. tabulaeformis* tree–ring sample (LCM0118A). The distinct earlywood (EW) and latewood (LW) segments can be identified by inspection under a microscope.

删除了: Scanned p

[Figure]

**Figure S2.** Standard (STD) tree-ring width chronologies (solid curves) generatued using three kinds of detrending methods for **(a–c)** earlywood width (EWW), **(d–f)** latewood width (LWW), and **(g–i)** total tree–ring width (TRW) at the two study sites, Baiyunshan (black) and Longchiman (red). The three kinds of detrending methods are: (1) negative exponential functions combined with linear regression with negative (or zero) slope (NELR), (2) cubic smoothing splines with a 50 % frequency cutoff of 67 % of the series length (SP67), and (3) age-dependent splines with an initial stiffness of 50 years (SPA50). The dashed and dotted curves denote the running expressed population signal (EPS) and effective chronology signal (Rbar$_{eff}$) , respectively. The horizontal line indicates the threshold EPS value of 0.85. The running EPS and Rbar$_{eff}$ values were calculated over a 51–year window.

删除了: together

删除了: ed

[Figure]

**Figure S3.** Signal-free (SSF) tree-ring width chronologies (solid curves) generatued using three kinds of detrending methods for **(a–c)** earlywood width (EWW), **(d–f)** latewood width (LWW), and **(g–i)** total tree–ring width (TRW) at the two study sites, Baiyunshan (black) and Longchiman (red). The three kinds of detrending methods are: (1) negative exponential functions combined with linear regression with negative (or zero) slope (NELR), (2) cubic smoothing splines with a 50 % frequency cutoff of 67 % of the series length (SP67), and (3) age-dependent splines with an initial stiffness of 50 years (SPA50). The dashed and dotted curves denote the running expressed population signal (EPS) and effective chronology signal (Rbar$_{eff}$) , respectively. The horizontal line indicates the threshold EPS value of 0.85. The running EPS and Rbar$_{eff}$ values were calculated over a 51–year window.

删除了: The same as Figure S2, but for the signal-free (SSF) chronologies. .
————————————————————分页符————————————————————

[Figure]

**Figure S4.** Composite **(a–c)** STD and **(d–f)** SSF tree-ring width chronologies for EWW (red), LWW (black), and TRW (blue) generated using the merged tree–ring samples from the two study sites, Baiyunshan and Longchiman, based on three kinds of detrending methods. The detrending methods are: (1) negative exponential functions combined with linear regression with negative (or zero) slope (NELR), (2) cubic smoothing splines with a 50 % frequency cutoff of 67 % of the series length (SP67), and (3) age-dependent splines with an initial stiffness of 50 years (SPA50). The dashed and dotted curves denote the running expressed population signal (EPS) and effective chronology signal (Rbar$_{eff}$), respectively. The horizontal line indicates the threshold EPS value of 0.85. The running EPS and Rbar$_{eff}$ values were calculated over a 51–year window. The segement plot indicated the sample depth (core).

删除了: together

删除了: ed

[Figure]

**Figure S5.** Spatial distribution of the meteorological stations included in the CRU dataset (http://www.cru.uea.ac.uk/data) around the tree-ring sampling sites, Baiyunshan and Longchiman. Black cycles represent the meteorological stations that provide both precipitation and temperature data. While, Blue (yellow) cycles represent those that only provide precipitation (temperature) data. The dashed rectangle indicates the range of scPDSI grid points used for calibration in this study. The color patches denote the spatial correlation coefficients between May-July scPDSI and NELR based EWW STD chronology.

删除了: blue

[Figure]

[Figure]

**Figure S6.** Correlation coefficients between the tree-ring width chronologies and multi-month averaged scPDSI (April to September of the current year). Upward triangles, downward triangles, and circles indicate the chronologies of EWW, LWW, and TRW, respectively. The color black, red and blue indicate the STD chronologies generated using the detrending methods NELR, SP67, and SPA50, while colors magenta, cyan, and orange indicate the SSF chronologies generated using the detrending methods NELR, SP67, and SPA50, respectively. The arrow indicates that the corresponding correlation does not reach the 0.05 significance level, which was tested using the Monte Carlo method (Efron and Tibshirani, 1986; Macias-Fauria et al., 2012).

[Figure]

**Figure S7.** Comparisons between the MJJ scPDSI (gray) and EWW STD chronologies generated using

the detrending methods NELR (black), SP67 (red), and SPA50 (blue) during the period 1953–2005. The

corresponding linear trends are indicated by the dashed lines, and the slope statistics are labelled in the

bottom left corner of the figure.